**Towards an assessment of riverine dissolved organic carbon in surface waters of the Western**
**Arctic Ocean based on remote sensing and biogeochemical modeling**
[1]Vincent Le Fouest
[2,3]Atsushi Matsuoka
[4]Manfredi Manizza
[1]Mona Shernetsky
[5]Bruno Tremblay
[2,3]Marcel Babin
[1]Littoral Environnement et Sociétés, UMR 7266, Université de La Rochelle, La Rochelle, France
[2]Takuvik Joint International Laboratory, Université Laval & CNRS, Québec, QC, G1V 0A6,
Canada
[3]Takuvik Joint International Laboratory, CNRS, Québec, QC, G1V 0A6, Canada
[4]Geosciences Research Division, Scripps Institution of Oceanography, University of California San
Diego, La Jolla, CA 92093-0244, USA
[5]Department of Atmospheric and Oceanic Sciences, McGill University, Montreal, QC, H3A OB9,
Canada

**Abstract**

Future climate warming of the Arctic could potentially enhance the load of terrigenous dissolved organic carbon (tDOC) of Arctic rivers due to increased carbon mobilization within watersheds. A greater flux of tDOC might impact the biogeochemical processes of the coastal Arctic Ocean (AO) and ultimately its capacity of absorbing atmospheric $CO_2$. In this study, we show that sea surface tDOC concentrations simulated by a physical-biogeochemical coupled model in the Canadian Beaufort Sea for 2003-2011 compare favorably with estimates retrieved by satellite imagery. Our results suggest that, over spring-summer, tDOC of riverine origin contributes to 35 % of primary production and that an equivalent of ~10 % of tDOC is exported westwards with the potential for fueling the biological production of the eastern Alaskan nearshore waters. The combination of model and satellite data provide promising results to extend this work to the entire AO so as to quantify, in conjunction with in-situ data, the expected changes in tDOC fluxes and their potential impact on the AO biogeochemistry at basin scale.

## 1. Introduction

The Arctic Ocean (AO) receives ~10% of the global freshwater discharge (Opsahl et al., 1999 and references therein), of which the larger part (~54-64 %) originates from six main pan-Arctic rivers (Haine et al., 2015; Holmes et al., 2012; Aagaard and Carmack, 1989). Over the past 30 years, the Arctic freshwater cycle intensified as reflected by changes in snow cover (Bring et al., 2016), evapotranspiration from terrestrial vegetation (Bring et al., 2016), and precipitation (Vihma et al., 2016). It resulted in an increase of the freshwater discharge from North American and Eurasian rivers by ~2.6 % and ~3.1 % per decade, respectively (Holmes et al., 2015). More than half the soil organic carbon stock on Earth is contained in the permafrost of the Arctic watersheds (Tarnocai et al., 2009). With the warming of the lower atmosphere, the permafrost undergoes a substantial thawing (Romanovsky et al., 2010) likely to alter the organic carbon content and quality of inland waters. In the past decades, the flux of dissolved organic carbon (DOC) decreased in the Yukon River (40 %; Striegl et al., 2005) while it increased at the Mackenzie River mouth (~39 %; Tank et al., 2016). These contrasting responses to climate change suggest that the direction of future trends of DOC concentrations and fluxes to the AO are very uncertain (Abbott et al., 2016).

The coastal AO influenced by large river plumes is hence exposed to changing conditions. Coastal waters are supplied in riverine organic carbon all year round with a maximal flux in spring-early summer when the freshwater discharge reaches a seasonal maximum. In river waters, DOC is present in higher concentration than the particulate form (Le Fouest et al., 2013; Dittmar et al., 2003). It accounts for more ~82 % of the flux of total riverine organic carbon (McGuire et al., 2009). The pan-Arctic flux of riverine DOC to the AO is estimated to be 33-37.7 TgC yr$^{-1}$ (Holmes et al., 2012; Manizza et al., 2009; McGuire et al., 2009; Raymond et al., 2007). As the organic carbon formed by phytoplankton, terrigenous DOC (tDOC) can be considered new carbon fueling annually the upper AO. In that respect, and regardless of its distinct nature and fate, the flux of riverine DOC would be equivalent to 10-19 % of AO primary production (Stein and Macdonald, 2004; Bélanger et al., 2013). In the oligotrophic Beaufort Sea, this proportion would reach ~34 % (S. Bélanger, pers.

comm.). Riverine DOC is hence a significant pool in the Arctic carbon cycle that can markedly modify the biological production and biogeochemistry of the AO waters. Within the pelagic food web, riverine DOC can be assimilated and transformed, promoting both phytoplankton and bacterioplankton production (Le Fouest et al., 2015; Tank et al., 2012). Riverine DOC can also modulate the air-sea fluxes of $CO_2$. In present climatic conditions, Manizza et al. (2011) suggest that the mineralization of riverine DOC into dissolved inorganic carbon would induce a 10 % decrease of the net oceanic $CO_2$ uptake at the pan-Arctic scale. On East Siberian shelves, the degradation of terrestrial organic carbon would be partly responsible for sea surface acidification (Semiletov et al., 2016).

In recent studies, riverine DOC flux data were used in a 3D ocean-biogeochemical coupled model to investigate the fate of riverine DOC within surface Arctic waters (Le Fouest et al., 2015; Manizza et al., 2013, 2011, 2009). However, simulated spatial and temporal changes in riverine DOC concentrations have not yet been compared with remote sensing data to assess the model predictive ability. Such a model-satellite comparison allows validating the model and then using it with confidence to resolve the annual cycle of riverine DOC, a prerequisite for a robust assessment of the riverine DOC contribution to the Arctic carbon cycle. To this end, riverine DOC concentrations at the sea surface obtained from a previous model run described in Le Fouest et al. (2015) and tDOC concentrations derived from remote sensing data were analyzed for the Canadian Beaufort Sea. As riverine DOC accounts for more than 99 % of the total tDOC exported to the AO (McGuire et al., 2009), we will use the term tDOC for both the model and remotely sensed data. Our goals are to compare tDOC data derived from the model and from remote sensing using skill metrics, in order to assess the model capacity to reproduce the observed seasonal and spatial variability in tDOC, and to provide bulk estimates of the seasonal tDOC stock and lateral fluxes within the surface coastal waters using a combination of these two approaches.

The paper is organized as follows. First, we describe the two different approaches used to quantify tDOC within the AO, i.e. a semi-analytical method based on remote sensing and a regional ocean-

biogeochemical coupled model that includes explicit fluxes of riverine DOC to the AO. Second, we compare the distribution and export flux of tDOC within surface waters of the Beaufort Sea estimated by the model and remote sensing. Finally, we discuss future developments of biogeochemical models necessary to simulate successfully the carbon budget of Arctic coastal waters in a warming world.

## 2. Material and methods

### 2.1 Remote sensing data

Level 1A scene images acquired from the MODerate-resolution Imaging Spectroradiometer (MODIS) aboard the Aqua satellite were downloaded from the NASA ocean color website (https://oceandata.sci.gsfc.nasa.gov/MODIS-Aqua/L1/). After geometric correction, remote sensing reflectance, $Rrs(\lambda)$ data at 412, 443, 488, 531, 555, and 667 nm were obtained by applying the atmospheric correction proposed by Wang and Shi (2009) with modifications adapted to Arctic environments (Doxaran et al., 2015; Matsuoka et al., 2016). The light absorption coefficients of colored dissolved organic matter at 443 nm ($a_{CDOM}(443)$) were derived from the $Rrs(\lambda)$ data using the gsmA algorithm (Matsuoka et al., 2017) that optimizes the difference between satellite $Rrs(\lambda)$ and $Rrs(\lambda)$ calculated using parameterization of absorption and backscattering coefficients for Arctic waters (Matsuoka et al., 2011, 2013). tDOC concentrations were estimated from the $a_{CDOM}(443)$ data using an empirical relationship between DOC and $a_{CDOM}(443)$ established in the Southern Beaufort Sea (Matsuoka et al., 2013). Since DOC concentrations estimated using ocean color data are based on a highly significant DOC versus $a_{CDOM}(443)$ relationship ($R^2 = 0.97$; Matsuoka et al., 2012), the DOC is considered to be of terrestrial origin. Errors of intercept, slope, and $a_{CDOM}(443)$ were propagated into the in-situ (empirical) DOC versus $a_{CDOM}(443)$ relationship. It resulted into a mean uncertainty of the tDOC concentration estimates of 28 % (see Appendix A2 of Matsuoka et al., 2017). Scene images of tDOC concentrations were used to make monthly

composite images at 1 km horizontal resolution of the Mackenzie shelf in the Canadian Beaufort
Sea (Fig. 1).

**2.2 3D physical-biogeochemical model data**
We used sea surface tDOC concentrations and ocean currents simulated over 2003-2011 by a
previous pan-Arctic model run ("RIV run") whose setup is fully detailed in Le Fouest et al. (2015).
The pan-Arctic model data were extracted on the remote sensing geographical domain focused on
the southern Beaufort Sea. We provide here a brief description of the physical-biogeochemical
coupled model used to generate the "RIV run". The MITgcm (MIT general circulation model)
ocean-sea ice model (Nguyen et al., 2011, 2009; Losch et al., 2010; Condron et al., 2009) has a
variable horizontal resolution of ~18 km and covers the Arctic domain with open boundaries at
55°N on the Atlantic Ocean and Pacific Ocean sides. The open ocean boundaries are constrained by
potential temperature, salinity, flow, and sea-surface elevation derived from integrations of a global
configuration of the MITgcm model (Menemenlis et al., 2005). Atmospheric forcings (10 m winds,
2 m air temperature and humidity, and downward long and short-wave radiation) are taken from the
six-hourly data sets of the Japanese 25 year ReAnalysis (JRA-25) (Onogi et al., 2007). In addition
to precipitations, the hydrologic forcing includes a monthly climatology of freshwater discharge
from 10 pan-arctic watersheds (Manizza et al., 2009). Monthly mean estuarine fluxes of freshwater
are based on an Arctic Runoff database (Lammers et al., 2001; Shiklomanov et al., 2000). For each
watershed, the river discharge forcing is associated with a monthly climatology of riverine DOC
concentration (Manizza et al., 2009). The total annual load of tDOC in the model is 37.7 TgC yr$^{-1}$. It
is consistent with previous values reported in Raymond et al. (36 TgC yr$^{-1}$; 2007) and Holmes et al.
(34 TgC yr$^{-1}$; 2012) and obtained by using load estimation models linking riverine DOC
concentrations to river discharge data. The physical model is coupled with a 10-compartment
biogeochemical model (Lee et al., 2016; Le Fouest et al, 2015). The biogeochemical model
explicitly accounts for dissolved inorganic nutrients (nitrate and ammonium), small and large
phytoplankton, protozooplankton, mesozooplankton, bacterioplankton, detrital particulate and
dissolved organic nitrogen, and tDOC (Lee et al., 2016; Le Fouest et al., 2015). The tDOC
compartment couples the marine and terrestrial cycling of organic matter though tDOC recycling
into inorganic nutrients by bacterioplankton. We set to 15 % the percentage of tDOC entering the
model as usable by the bacterioplankton compartment. This value was estimated based on the mean
yearly percentages of the total load of riverine DOC considered as biodegradable DOC for six
major Arctic rivers given in Wickland et al. (2012).

**2.3 Analysis**
Remotely sensed and simulated tDOC data were binned for the months of June, July, August and
September over the 9-year period (2003-2011) to get the best areal coverage in the satellite
composites. The remotely sensed tDOC concentrations were regridded on the model horizontal grid.
Skill metrics were used to compare the remotely sensed estimates of tDOC with their simulated
counterparts. The metrics included the correlation coefficient (r), the unbiased root mean square
error (RMSE), the Nash-Sutcliffe model efficiency index (MEF), the geometric bias, and the
geometric RMSE (see Stow et al., 2009; Doney et al., 2009; Nash and Sutcliffe, 1970). The metrics
are computed as follows:

$$r = \frac{\sum_{n=1}^{N}(sat_n -)(mod_n - \overline{mod})}{\sqrt{\sum_{n=1}^{N}(sat_n - \overline{sat})^2 \sum_{n=1}^{N} mod_n - \overline{mod}^2}} \qquad (Eq.\,1)$$

$$unbiased\ RMSE = \sqrt{\frac{1}{N}\sum_{n=1}^{N}\left(mod_n - sat_n - (\overline{mod} - \overline{sat})\right)^2} \qquad (Eq.\,2)$$

$$MEF = \frac{\sum_{n=1}^{N}(sat_n - \overline{sat})^2 - \sum_{n=1}^{N}(sat_n - mod_n)^2}{\sum_{n=1}^{N}(sat_n - \overline{sat})^2} \qquad (Eq.\,3)$$

$$geometric\ bias = e^{(\overline{mod} - \overline{sat})} \qquad (Eq.\,4)$$

$$geometric\ RMSE = \sqrt{e^{\left(\frac{1}{N}\Sigma_{n=1}^{N}(mod_n - sat_n)^2\right)}} \qquad\qquad (Eq.5)$$


where N is the number of tDOC data, and $\overline{sat}$ and $\overline{mod}$ are the remotely sensed and the simulated
tDOC averages, respectively. Monthly fluxes of tDOC were calculated and summed along two
cross-shelf transects (see upper-middle panel in Fig. 2). At each grid cell, the model flux estimate
was computed as the product of the simulated sea surface current velocity with the simulated tDOC
concentration. The remote sensing flux estimate was computed as the product of the simulated sea
surface current velocity with the remotely sensed tDOC concentration.

**3. Results and discussion**
**3.1 tDOC concentrations and distribution**
Over the Mackenzie shelf, the plume of high-tDOC ($> 120$ mmolC m$^{-3}$) had a maximal areal extent
in June for both the model and the satellite data (Fig. 2). This coincided with the seasonal peak of
river discharge in June as parameterized in the model and generally depicted by in-situ time series
(Yang et al., 2015). From July to September, the high-tDOC areal extent progressively decreased
following the seasonal pattern of riverine freshwater discharge (see Yang et al., 2015; Manizza et al.,
2009). This seasonal pattern was observed both in the model and satellite data. The simulated tDOC
concentrations were lower than in the satellite record in Mackenzie Bay and east of the Mackenzie
Bay, especially in June (by 44 % in average) and July (by 27 % in average). In the Beaufort and
Chukchi seas, first year sea ice represents a carbon flux to the ocean of $2 \times 10^{-4}$ TgC yr$^{-1}$ (Rachold
et al., 2004). This flux is 4 orders of magnitude lower than the tDOC supply from the Mackenzie
River specified as boundary conditions in the model (2.54 TgC yr$^{-1}$). Similarly, tDOC eroded from
permafrost stored in the North American shores would account for only $\sim 0.5$-$1.6 \times 10^{-4}$ TgC yr$^{-1}$
(Tanski et al., 2016; Ping et al., 2011, using a DOC:POC ratio of 1:900 as in Tanski et al., 2016) to
$\sim 2 \times 10^{-3}$ TgC yr$^{-1}$ (McGuire et al., 2009). With regard to these flux values, tDOC originating from

both melted sea ice and eroded permafrost, not taken into account in the model, are hence not believed to explain the model-satellite discrepancies (Fig. 2). Other factors might contribute to these model-satellite differences observed nearshore. First, the model does not distinguish between the two main pathways of the Mackenzie River discharge entering the shallow delta zone. In June, the Mackenzie Bay receives most of the fresh and turbid river water (~66 %) while the remaining ~33 % spreads east of the delta in Kugmallit Bay (Davies, 1975). This pattern was particularly well captured by the remotely sensed data in June-July (Fig. 2). Second, the inner Mackenzie shelf (< 20 m depth) is bounded during winter by a thick ridged ice barrier grounded on the sea floor called stamukhi (Macdonald et al., 1995). The stamukhi retains the turbid river water within the inner shelf in winter. When sea ice breaks up and the freshet reaches its seasonal maximum in spring, the retained turbid waters spread farther within the coastal zone. Contrary to the model, the remote sensing data could resolve this particular feature explaining the higher tDOC concentrations observed nearshore in June (see Fig. 2). Such a pattern observed for tDOC is also reported for terrigenous particulate organic matter (Doxaran et al., 2015). Further offshore on the Mackenzie shelf, as delimited by the 300 m isobaths both remotely sensed and simulated concentrations of tDOC were within the range of values measured in spring (~110-230 mmolC $m^{-3}$; Osburn et al., 2009) and summer (~60-100 mmolC $m^{-3}$; Para et al., 2014). The simulated values of tDOC were higher than those remotely sensed on the outer and off the shelf. Overall, the model and the satellite data captured the seasonal cycle and spatial distribution of tDOC concentrations in the study area.

Skill metrics were computed over the whole study area (see Fig. 2) to provide a quantitative comparison of tDOC simulated with the model and satellite data (Table 1). For all months, the correlation coefficient was relatively high (0.78<r<0.82) within the range of values obtained for sea surface dissolved inorganic nutrients simulated by global models (r>0.75; Doney et al., 2009). Regardless of amplitude, the r values showed that the simulated and remotely sensed tDOC concentrations presented similar patterns of variation. The size of the model-satellite discrepancies was given by the unbiased RMSE. Overall, the unbiased RMSE decreased from June (41.4 mmolC

m$^{-3}$) to September (29.3 mmolC m$^{-3}$). This result suggested that the model accuracy increased from
spring to summer. The model capability for predicting tDOC relative to the average of the remote
sensing counterparts was estimated by the model efficiency index (- ∞ <MEF ≤ 1) (Nash and
Sutcliffe, 1970). The MEF is a normalized statistic that relates the residual variance between the
simulated and remotely sensed tDOC concentrations to the variance within the remotely sensed
tDOC data (see Eq. 3). A MEF value near zero means that the residual variance compares to the
remotely sensed variance, i.e. that the model predictions are as accurate as the mean of the satellite
data. As the MEF increases towards a value of one, the residual variance becomes increasingly
lower than the observed variance. For all months, the MEF was positive (0.26-0.60) suggesting that
tDOC concentrations simulated by the model were an acceptable predictor relative to tDOC
concentrations derived from remote sensing, especially in June-July. In order to give a more even
weight to all of the data and to limit the skewness towards the higher tDOC concentrations, metrics
based on log-transformed tDOC data were also computed. For all months, the geometric RMSE was
close to one and range between 1.02 and 1.12. It suggested that the model-satellite data dispersion
was relatively small when the positive skewness was reduced. In June, the relatively high unbiased
RMSE could be partly due to high tDOC concentrations as suggested by the relatively low
geometric RMSE (1.07). Finally, the computed geometric bias informs with respect to the direction
of the model-satellite discrepancies. For all months, the geometric bias (1.07-1.32) was higher than
one meaning that the model tended, on average, to overestimate the observations over the whole
domain. The highest geometric bias was reported in August (1.32), when the river discharge was
low, suggesting that tDOC removal was likely underestimated in the model in late summer. A
Taylor diagram (Taylor, 2001) was produced to provide a synthetic and complementary overview of
how the simulated and remotely sensed tDOC concentrations compared seasonally in terms of
correlation, amplitude of variations (given by the standard deviations), and normalized model-
satellite discrepancies (Fig. 3). All months differed by their normalized RMSE and amplitude of
variations while the correlation coefficient was close to ~0.8 (see Table 1). The model best
performed in simulating tDOC in July, just after the seasonal peak of river discharge, followed by
the months of June and August. June and August showed similar values of correlation, RMSE, and
normalized standard deviation despite distinct seasonal patterns of river discharge (high and low,
respectively). By contrast, September showed the highest model-satellite data dispersion. With
respect to satellite estimates, the skill metrics overall suggested that the model could reliably
simulate tDOC concentrations in surface waters over a wide range of river discharge and tDOC load.

**3.2 tDOC stock and lateral export fluxes**
The overall agreement between the model and the satellite tDOC concentrations allowed the
assessment of the mean areal stock and lateral fluxes of tDOC using the mean surface ocean
circulation simulated by the MITgcm (Table 2). The monthly-averaged (June to September) areal
stock of tDOC over the Mackenzie shelf as delimited by the 300 m isobaths was estimated to 1.37
TgC (Table 2). The bias between the model and the satellite data was the highest in August but did
not exceed +8.2 % (0.1 Tg C). This result is consistent with the highest geometric bias reported in
August (Table 1). In the model, the removal of tDOC through photo-oxidation (Bélanger et al.,
2006) was not taken into account. Assuming an annual mean mineralization rate of tDOC of ~0.02
TgC (Bélanger et al., 2006), this process would explain <2 % of the reported tDOC difference in
August. In addition, the 15% value used to set the bioavailable tDOC fraction in the model was at
the low end of values reported in other studies (up to 50%; Mann et al., 2012; Wickland et al., 2012,
Letscher et al., 2011; Alling et al., 2010; Holmes et al., 2008). This underestimation of the
bioavailable fraction of tDOC upon delivery to the AO could be a major reason why the simulated
values of tDOC were consistently overestimated when compared to satellite estimates for the outer
shelf and offshore locations (Fig. 1, Table 1). In the model, bacterioplankton consumed tDOC to
produce ammonium usable in turn by phytoplankton. In the Beaufort Sea, this pathway contributed
to primary production by 35 % on average over 2003-2011. However, the simulated rates of
bacterioplankton production ($< 30$ mgC m$^{-2}$ d$^{-1}$) still remained in the lower range of those measured
in the Beaufort Sea (25-68 mgC $m^{-2}$ $d^{-1}$; Ortega-Retertua et al., 2012; Vallières et al., 2008). The
likely underestimation of the tDOC removal by bacterioplankton in the model during summer
months might largely contribute to the reported bias between the model and the satellite data.
Nevertheless, the bias remained moderate with respect to values reported for June, July and
September (-1.5 % to -2.8 %) (Table 2).
Combining the modeling and remote sensing approaches allowed for the reconstruction of the
dominant surface pattern in lateral tDOC fluxes in the Canadian Beaufort Sea from June to
September (Fig. 4). Two north-south transects were defined east (Cape Bathurst) and west
(Mackenzie Trough) of the Mackenzie shelf (see upper-middle panel in Fig. 2). The net seasonal
flux was westward along the two transects following the anticyclonic circulation pattern of the
Beaufort gyre (Mulligan et al., 2010) and was maximum in June and September. The flux was at
least three times higher along the western transect near the Mackenzie Through than east at Cape
Bathurst. This suggests a net export of tDOC towards the Alaskan part of the Beaufort Sea. In
contrast, whilst the flux in July and August remained oriented westward near the Mackenzie Trough,
it was reversed at Cape Bathurst. In July, the tDOC flux was still 1.3 to 1.7 times higher along the
western transect. In August, however, there was more tDOC (~1.4-fold) exported eastward at Cape
Bathurst than exported westward near the Mackenzie Through.
Along the two transects, the simulated fluxes were higher than those derived from remotely sensed
tDOC concentrations (Fig. 4). The monthly bias between the model and the satellite flux estimates
varied between 0 % and +18.2 %. The bias on the seasonal net flux was moderate (+8.3 %) near the
Mackenzie Trough but reached +25 % at Cape Bathurst. The seasonal mean flux however was one
order of magnitude lower than near the Mackenzie Trough. The flux estimates suggested that,
despite discrepancies in tDOC concentrations, the modeling and remote sensing approaches
provided robust estimates of the lateral transport of tDOC in surface waters in late spring-summer.
Because of sea ice and cloud cover, the satellite retrieval was limited to a temporal window
covering a third of a year only, i.e. from June to September. The yearly mean lateral flux of tDOC

was computed from the simulated data along the Mackenzie Trough transect and it reached 0.31 TgC. The flux of tDOC cumulated over June to September along this transect (0.12-0.13 TgC) represented ~42 % of this annual flux (0.31 TgC), which is consistent with the fraction of the annual discharge of freshwater by the Mackenzie that occurs during spring-summer (~50 %; McClelland et al., 2012). Using stable isotope techniques on pelagic particulate organic matter, Bell et al. (2016) showed that OC originating from the Mackenzie outflow in summer was incorporated within bentho-pelagic food webs as far as the eastern Alaskan shelf. In nearshore waters of this part of the Beaufort Sea, the study of Dunton et al. (2006) using stable isotopes also suggested that tDOC from the Mackenzie River could add to the local terrigenous carbon inputs mediated by coastal erosion and smaller rivers to fuel the biological production in summer. Using the model and satellite data, we report that an equivalent of ~10 % (0.12-0.13 TgC) of the cumulated flux of tDOC delivered by the Mackenzie River over spring-summer (1.32 TgC) was exported westward in the Alaskan Beaufort Sea along the Mackenzie Trough transect.

## 4. Perspectives

The results of our study suggest that the model is in fair agreement with the surface tDOC fields remotely sensed in spring-summer when most of the riverine flux occurs. The comparison allows an evaluation of the model and justifies its use to resolve the annual cycle of tDOC. Because satellite imagery provides data only during spring-summer, further uncertainties still remain in the model in fall-winter in terms of tDOC concentrations and spatial distribution. In addition, the model involves some limitations mostly due to the biogeochemical processing of tDOC. The tDOC transformation is complex to translate into robust mechanistic equations as highly dependent on the availability of in-situ data in Arctic waters. For instance, the riverine tDOC compartment is split in the model into a labile and a non-labile fraction (see Le Fouest et al., 2015). This parameterization strongly constrains the removal of tDOC by bacterioplankton and therefore the tDOC concentrations simulated within surface waters. In natural waters, however, tDOC is made of a complex mixture of

compounds that differ by their chemical composition and age (Mann et al., 2016) and so along the seasons (Wickland et al., 2012, Mann et al., 2012). The chemical nature of tDOC impacts its bioavailability, which is estimated to average 6 % to 46 % of the total tDOC pool with marked disparities amongst the seasons and the rivers (Mann et al., 2012). Nevertheless, the general trend for the six major Arctic rivers (Kolyma, Yukon, Mackenzie, Ob, Yenisey and Lena) is a more labile tDOC pool in winter than in spring and summer (Wickland et al., 2012). In the Kolyma River, Mann et al. (2012) report a higher labile fraction in spring (~20 %) than in summer (<10 %) as the exported tDOC is younger during the freshet. Such a pattern is, however, not clearly present in the Mackenzie River (e.g. Wickland et al., 2012). We suggest that a more realistic representation in the model of the nature of the organic matter entering the coastal waters might improve the tDOC concentrations simulated in surface AO waters. It could include, for instance, the riverine flux of both dissolved organic carbon and nitrogen along with an improved C:N stoichiometry for bacterioplankton uptake (see Le Fouest et al., 2015).

In the model, the seasonal forcing of tDOC was based on DOC measurements gathered hundreds kilometers upstream the rivers' mouths. This precludes any DOC enrichment of the Mackenzie River water as it flows through the delta (see Emmerton et al., 2008) with, as a consequence, a likely underestimation of tDOC concentrations simulated in nearshore waters. Therefore, the quantification of the tDOC flux from the watersheds to the coastal AO poses as another key issue to addressing the role of tDOC in the biogeochemistry of shelf waters. Recently, watersheds models were developed to assess this tDOC flux (Tank et al., 2016; Kicklighter et al., 2013; Holmes et al., 2012). Such models provide realistic estimates but still require improvements as watersheds properties and mechanistic processes underlying the tDOC mobilization and riverine transport are complex to set up (see Kicklighter et al., 2013). The remote sensing of high resolution ocean color data is increasingly used to assess tDOC concentrations in large pan-Arctic rivers during the open water season (Herrault et al., 2016; Griffin et al., 2011). Ocean color techniques could then prove

useful in the future to improve the tDOC time series set at models boundaries by accounting for
instance for year-to-year variations of tDOC concentrations during the freshet period.
In our study, the remotely sensed tDOC concentrations retrieved in shelf waters provide the
advantage of already integrating the effect of the watersheds processes such as mobilization,
transformation and transport at the seasonal and synoptic time scales. However, we acknowledge
that the temporal coverage of the remote sensing data is restricted to spring and summer. Because of
clouds and sea ice, we miss the winter season when tDOC is the most labile (e.g. Wickland et al.,
2012) and likely subject to remineralization. In the Mackenzie River, about 25 % of the annual load
of labile tDOC occurs during winter (Wickland et al., 2012). Despite this limitation, and in regard to
the model-satellite data comparison, the assimilation of remotely sensed tDOC data into Arctic
models could still offer an interesting perspective as it might result in more realistic simulated fields
of tDOC in spring and summer when the river discharge and tDOC export is the highest. Physical
and biological data have already been assimilated into Arctic predictive models to make the
simulated sea surface temperature, salinity, sea ice extent and thickness, and chlorophyll more
reliable (Simon et al., 2015; Massonnet et al., 2015). We may hence expect the assimilation of
remotely sensed tDOC concentrations to mitigate, at least partly, the issues linked to setting up
realistic tDOC forcings within predictive models. For instance, the assimilation of remotely sensed
tDOC data in open waters might help accounting for the interannual variations of tDOC delivered
by rivers, which are not resolved by the coupled model that is constrained by a monthly climatology
of tDOC load (see Manizza et al., 2009).
Improving the capability of Arctic models to resolve the fate and pathways of tDOC in the AO will
require certain limitations to be unlocked. To this purpose, future model developments must lie on
the always increasing observational effort realized by mean of field campaigns and new remote
sensing techniques. Observations must be used to improve the riverine forcings in order to better
encompass the seasonal to interannual variability of the terrigenous dissolved organic matter
exported to the coastal AO. Bacterioplankton dynamics also must be better represented in

biogeochemical models. In particular, the processes related to the competition for resources such as dissolved organic carbon and nitrogen of both allochtonous and autochtonous origin are likely to play an important role in mediating bacterioplankton growth and tDOC remineralization in Arctic coastal waters impacted by river plumes. Realistic fields of tDOC simulated by Arctic ocean-biogeochemical coupled models would be helpful for a more accurate assessment of $CO_2$ fluxes at the ocean-atmosphere interface. Arctic models that would combine realistic terrestrial fluxes of organic matter along with a robust representation of the pathways and processes responsible for its transformation in the AO would open an interesting perspective to address the effect on the Arctic carbon cycle of ongoing and future changes in the land-ocean continuum. The increase in seawater temperature of the AO due to global warming (Timmermans, 2016) might promote in the future the metabolism and respiration rates of marine bacterioplankton (Vaquer-Sunyer et al., 2010; Kritzberg et al., 2010). This enhanced microbial activity could then liberate extra nutrients provided by the remineralization of terrigenous organic matter that will then be available for primary production. This process might have an impact not only on the seasonal cycle of PP in the AO but also implications for the higher levels of the marine food webs of the AO, both benthic and pelagic.

**Data availability**

Data used in this study are available at http://www.obs-lienss.cnrs.fr/Publications/BGD_data_nc.tar.

**Acknowledgments**

This research was funded by the Centre national d'études spatiales (CNES) grant #131425-BC T23 to VLF and the Japan Aerospace Exploration Agency (JAXA) GCOM-C project through grant #16RSTK-007867 to AM. We thank a joint contribution to the research programs of UMI Takuvik (CNRS & Université Laval), ArcticNet (Network Centres of Excellence of Canada) and the Canada Excellence Research Chair in Remote Sensing of Canada's New Arctic Frontier (MB). We thank

Dimitris Menemenlis and the Estimation of Circulation and Climate of the Ocean (ECCO) group
from MIT for providing the physical model we used in this study. We also thank Cécilia Pignon-
Mussaud (LIENSs) for her help in processing the figure 1.


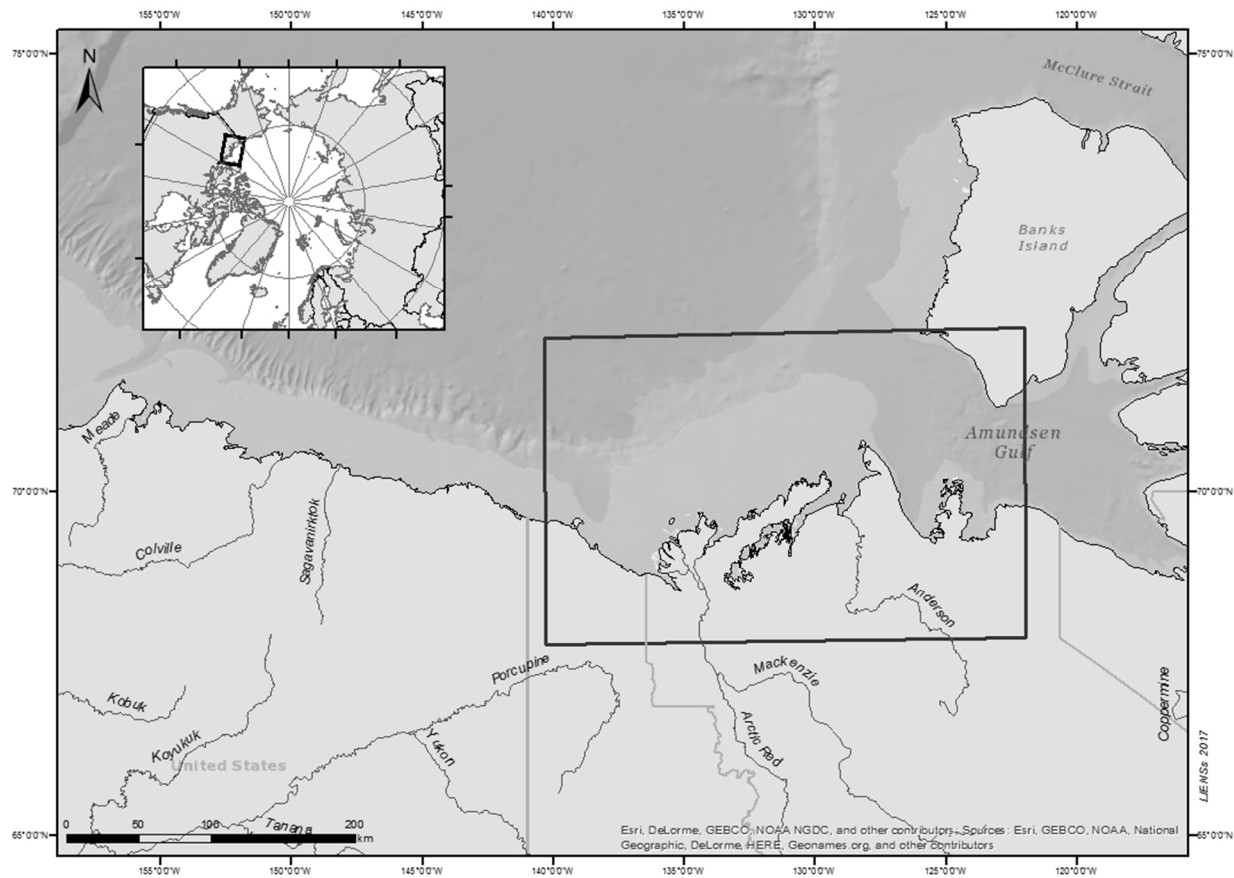

**Figure 1.** Map of the Canadian Beaufort Sea. The location of the study area is outlined with a

rectangle.


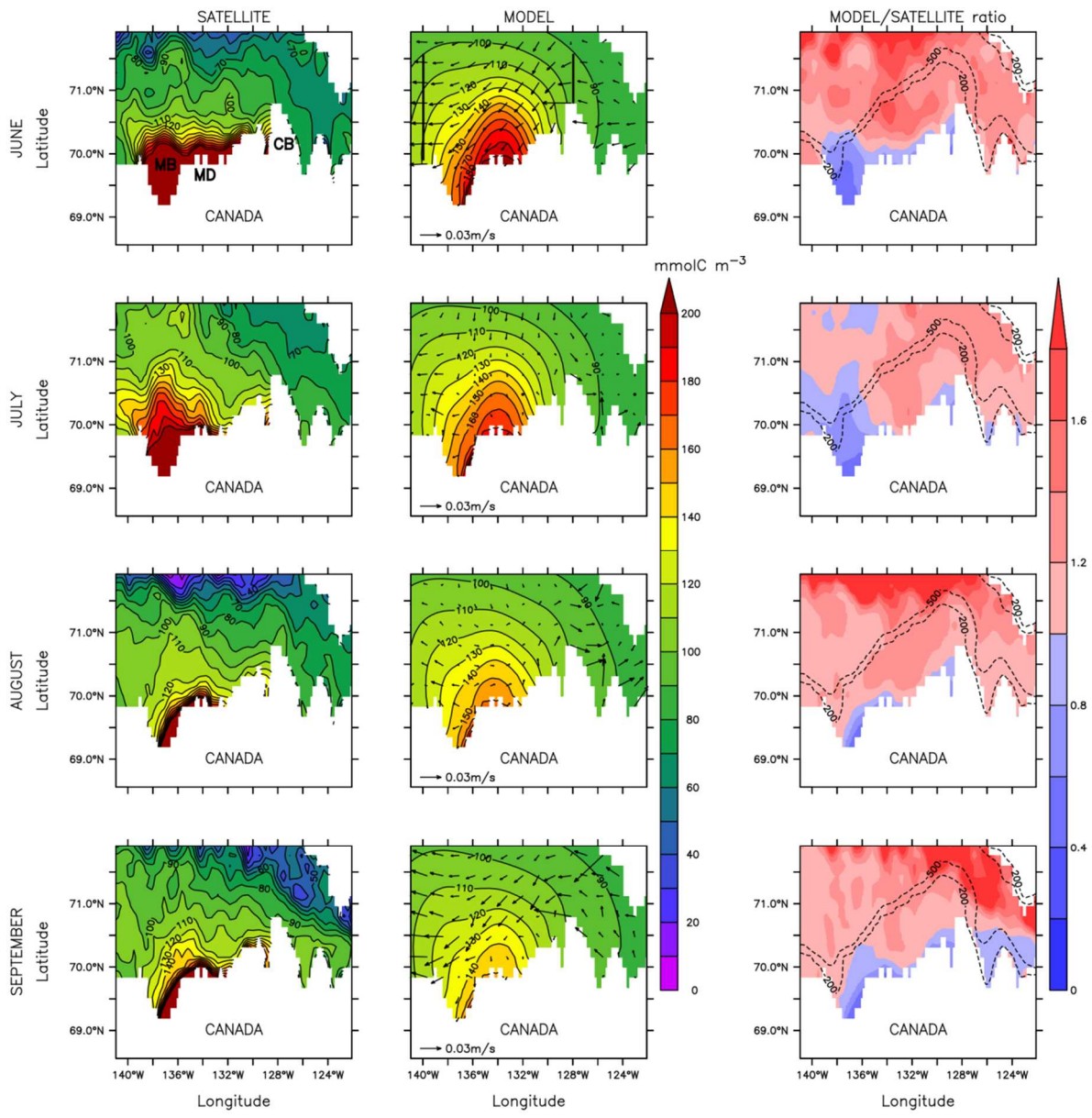


**Figure 2.** Monthly climatology (2003-2011) of surface tDOC concentration (mmolC m$^{-3}$) in the Beaufort Sea estimated from remotely sensed ocean color data (left panels) and by the biogeochemical model (middle panels) for June, July, August and September. The Mackenzie Bay (MB), Mackenzie delta (MD) and Cape Bathurst (CB) cited in the text are shown on the upper left panel. The isolines of tDOC concentration are overlaid (black full lines). In the middle panels, simulated surface currents (m s$^{-1}$) are overlaid. The two straight lines in the upper-middle panel refer to transects along which surface tDOC fluxes were computed. The right panels show the model over satellite tDOC data ratio with the 200 m and 500 m isobaths overlaid.

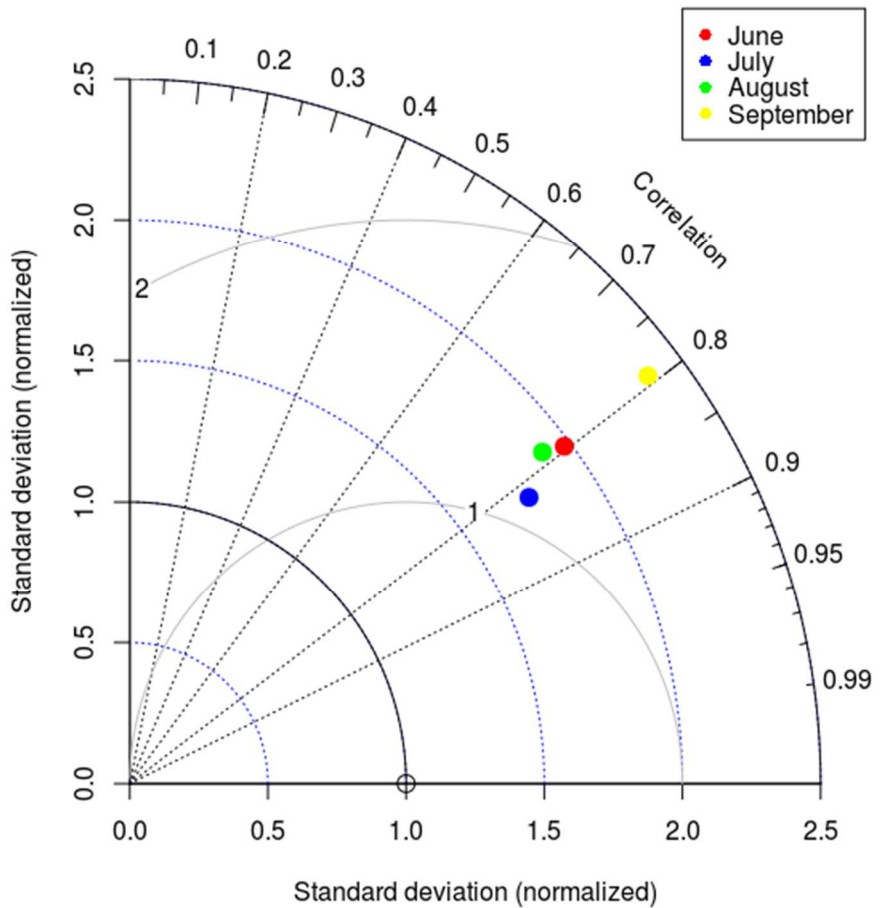


**Figure 3.** Taylor diagram displaying a statistical comparison between the simulated and remotely

sensed tDOC concentrations. The x-axis and y-axis show the model standard deviation relative to
the satellite standard deviation. The open circle on the x-axis represents the reference point. The
model-satellite correlation is represented in polar coordinates (angle from the x-axis). The light grey
full lines indicate the RMSE relative to the satellite standard deviation.

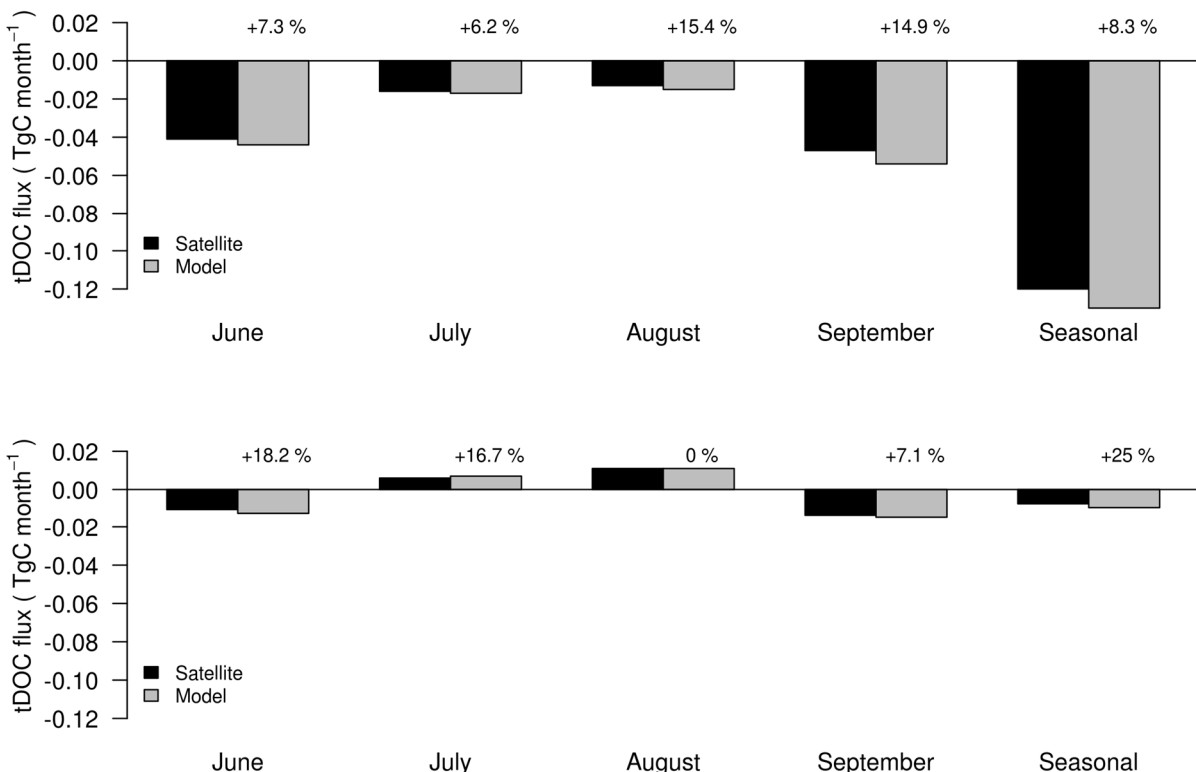

**Figure 4.** Monthly flux of surface tDOC (TgC month$^{-1}$) computed along transects located west of the Mackenzie Trough (139°W ; 69.5°N-71°N) (upper panel) and at Cape Bathurst (128°W ; 69.5°N-71°N) (lower panel). Transects are shown in figure 1 in the upper-middle panel. Negative values indicate a westward flux. Percentages refer to the model data relative to the satellite data. The seasonal flux refers to the 4-month net flux.

**Table 1.** Skill metrics of comparison computed based on the 2003-2011 monthly climatologies of
tDOC.

| Metric | June | July | August | September |
|---|---|---|---|---|
| Correlation coefficient | 0.79 | 0.82 | 0.78 | 0.79 |
| Unbiased RMSE (mmolC m$^{-3}$) | 41.4 | 29.4 | 26.0 | 29.3 |
| Model efficiency | 0.49 | 0.60 | 0.26 | 0.38 |
| | | | | |
| Geometric statistics using log-transformed data | | | | |
| Model bias | 1.24 | 1.07 | 1.32 | 1.21 |
| RMSE | 1.07 | 1.02 | 1.12 | 1.06 |



**Table 2.** Areal stock (TgC) of sea surface tDOC computed over the Mackenzie shelf (delimited by
the 300 m isobaths) from the model and satellite data. The bias (%) refers to the model data relative
to the satellite data. The seasonal areal stock refers to the 4-month average ± standard deviation.

|  | June | July | August | September | Seasonal |
|---|---|---|---|---|---|
| Model | 1.48 | 1.40 | 1.32 | 1.28 | 1.37±0.07 |
| Satellite | 1.51 | 1.44 | 1.22 | 1.30 | 1.37±0.11 |
| Bias | -2 | -2.8 | +8.2 | -1.5 | 0 |

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
