# Peer review of "1. Introduction"

_Biogeosciences, 2017_

## Referee Comment (RC1) · Anonymous Referee #1 · 11 Aug 2017

This manuscript is a comparison between satellite remote sensed estimates of DOC entrained with the Mackenzie River plume and biogeochemical model outputs. The authors find good agreement between the two estimates and suggest that additional insights could come from extending the work to the Arctic Ocean scale and in the context of climate warming that could be expected to affect DOC outflow from Arctic rivers.

The paper is authored by several of the prominent contributors to understanding dissolved organic carbon cycling in the Arctic, and is succinct, but I found the reasoning

circular as to the outcome of the study. Satellite remote sensing obviously can convey information on the directional flow of river plumes carrying DOC, but depth penetration from satellite platforms is modest, so without field sampling, comparison of one set of estimates with another produced by biogeochemical modeling seems like a limited and incomplete outcome. Moreover, many of the important areas of concern in the context of climate change revolve around the dynamics of DOC degradation. This process has higher rates in the spring freshet that later in the summer, and the different pools of marine and riverine DOC have different degrees of bioavailabilty. I didn't see this addressed significantly, including the extent to which DOC is removed in the river delta or near-shore zone, and after it is accounted for in flux estimates, but before it reaches the open ocean where estimates can be made from satellite platforms. It is also significant that much of the spring freshet flows over and under coastal sea ice from the Mackenzie River, but there is little inference about how that is accounted for. Comparisons are made to primary production, and it is stated that DOC from rivers represents 10-19% of the carbon fixed by primary production in the Arctic Ocean as a whole and up to 34% of primary production in the coastal Beaufort Sea, but the labile nature of organic carbon that is formed by marine production is quite different from most of the organic carbon in RDOC. It should be mentioned that the authors acknowledge some of these limitations in a general sense, including seasonal challenges to gathering satellite data, and the complex nature of RDOC in the Perspectives section, although those limitations are not reflected in the abstract of the study, which reads more optimistically.

The manuscript could be improved by light editing by a Native English language writer. Data supporting the study are available on-line, but no metadata or "read-me file" explaining use of the on-line files is provided. Ultimately, this manuscript is most appropriately seen as a limited follow-on to the Le Fouest et al. 2015 biogeochemical modeling paper, with the addition of a comparative approach to assessing satellite remote sensing data. I see no reason the manuscript couldn't be improved and accepted for publication, but I am skeptical of its potential for providing a more transformative understanding of dissolved organic carbon cycling in the Arctic.

---

## Referee Comment (RC2) · Anonymous Referee #2 · 29 Aug 2017

Towards an assessment of riverine dissolved organic matter in surface waters of the Western Arctic Ocean base on remote sensing and biogeochemical modeling

Overall: This study presents an interesting comparison of satellite-derived riverine DOC (RDOC) with biogeochemical model outputs. They find that estimates of RDOC in surface waters of the Canadian Beaufort Sea computed for 2003-2011 by both optical remote sensing and a physical biogeochemical mode compare favourably. Both display similar seasonal and spatial patterns in RDOC, with greater quantities of RDOC in June and a reduction in concentrations during summer to autumn months. How-

ever, some large differences in the absolute concentrations were discovered (e.g up to 44%). These results demonstrate the utility of validating model estimates using satellite -derived optical measurements of terrestrial OM flux during the summer. Overall, I found that the manuscript was a useful addition especially as it highlighted many of the potential current limitations with model and satellite approaches (e.g. adequately parameterising seasonal changes in lability, model estimates potentially not representing transformation and losses - esp. over summer months). I see the manuscript strengths as showing the direction of travel for this type of research, so as such I would like to see the later section about future directions to be strengthened, and more definitive suggestions provided. Further, it was not always clear to me what was new, or re-analysis of previously published research. This should be made more clear.

Specifics:- Introduction Line 27 - no need for thus. Line 31 - no need for riverine as implicit in RDOC. Line 32 - with *the * potential for fueling Line 35 - Awkward ending. Maybe consider something like: "Future studies could apply . . . the entire AO to quantify.. Line 39 - did you mean from the *six* great Arctic rivers in this paper? Line 40-41 - other factors contribute to the this intensification e.g. snow cover reduction, terrestrial productivity changes. Needs more detail here or suggest that increasing precipitation is one example. Line 41- grammar needs correcting Line 43 - contains half the soil *organic C stock Line 45 - maybe worth mentioning that it is currently unclear though if aquatic OC concentrations will increase, and that some studies suggest that OC concentrations may reduce (see Abbott et al 2016 and Striegl et al. 2005 for example) . Line 46 - not particularly suitable reference for the later part of the sentence re. changing OC concentration and composition. Suggest you use another and move the excellent Romanovsky one earlier in sentence. Line 49 - these rivers flow all year round, so OM supply does not only occur after the ice breakup period. Line 50 - seasonal in twice. Line 57 - unusual to have a pers comm here as well as the Manizza paper. Recommend removing as adds little evidence. Line 61 - can this be written more clearly. Its an important point, so how is RDOC reducing C uptake by 10%? Or is it offsetting this? Line 70 - so this is the same biogeochemical model

results from Le Fouest 2015? Please make this explicit here. What about the remote sensing component, is that new or also from previous work?

Materials and methods 90 - more details on the satellite products used and their source would be useful here. 93 - unclear grammar here so not sure how you are coming up with this uncertainty value. 97 - so are you including new model runs here or are they the same as subsequently published? 112 - please state how Raymond calculated this estimate. 119 - does Wickland really show this? I think she shows that between 12-18% of RDOC is available but that the average % is 15% in the Yukon river only. Please provide detail on assumptions. 136 - please reword this sentence for clarity.

Results & Discussion 146 - you define an acronym for simulated RDOC (RDOCsim) in the methods but then don't use it in this section. 148 - quite speculative this. Are you suggesting that this may account for the differences and can you justify this with any estimates? Most would not consider ice-derived plankton terrestrially derived also, so please re-phrase. 150 & 154 - should this read 2 x 10? Please update. 156 - ok so here you say this is not likely to be the cause. 157 (e.g. ??) 162 - less than 20 m of depth/ distance? 168 - Further offshore? 183 - I'm not clear on how this works? RMSE shows that the model was more 'accurate' after the spring flush. Yet, the MEF index shows that model and observations were closest during and just after the flush? Can you explain the discrepancy here, or am I misunderstanding? 184 - why does a positive MEF indicate this? 195 - please re-word to make this sentence clearer.

General Text could benefit from editing for English grammar. References are not in alphabetical order in places e.g. Raymond ref higher up etc. Is it appropriate to use RDOC as a term for the flux of C in the shelf region when it may be derived of a significant proportion of non riverine-derived OC?

Refs: Abbott, B. W., Jones, J. B., Schuur, E. A. G., Chapin, F. S., III, Bowden, W. B., Bret-Harte, M. S., et al. (2016). Biomass offsets little or none of permafrost carbon release from soils, streams, and wildfire: an expert assessment. Environmental

Research Letters, 11(3), 034014–14. http://doi.org/10.1088/1748-9326/11/3/034014

Striegl, R. G., Aiken, G. R., Dornblaser, M. M., Raymond, P. A., & Wickland, K. P. (2005). A decrease in discharge-normalized DOC export by the Yukon River during summer through autumn. Geophysical Research Letters, 32(21), L21413. http://doi.org/10.1029/2005GL024413

---

## Author Comment (AC1) · 4 Oct 2017

We gratefully thank referee #1 for her/his constructive comments with respect to our manuscript. In order to improve the manuscript with respect to these comments, we amended the manuscript as suggested by the referee wherever it was possible. Note that, when needed, comments were merged together to bring more clarity in the answer:

1. "Satellite remote sensing obviously can convey information on the directional flow

of river plumes carrying DOC, but depth penetration from satellite platforms is modest, so without field sampling, comparison of one set of estimates with another produced by biogeochemical modeling seems like a limited and incomplete outcome."

We agree with that comment in the sense that numerical modeling and remote sensing are not exhaustive approaches. Both are fully dependent from field measurements as their setup (e.g. forcings and differential equations for the model, algorithms for remote sensing) necessarily requires large in-situ databases. In our study, the model was constrained by riverine DOC observations at the boundaries of its numerical domain (see Manizza et al., 2009) while the semi-empirical remote sensing algorithm we used was developed based on field measurements (see Matsuoka et al., 2017). As explicitly mentioned in the manuscript, the modeling and remote sensing approaches combined together provide a relevant insight on the RDOC dynamics over a wide spatial and temporal scale, but limited to the surface coastal waters where RDOC concentrations are the highest.

In order to account for the referee's comment, we modified the end of the abstract (line 33) as follows: "Future studies could apply the combination of model and satellite data extended to the entire AO to quantify, in conjunction with in-situ data, the expected changes in RDOC fluxes and their potential impact on AO biogeochemistry."

2. "Moreover, many of the important areas of concern in the context of climate change revolve around the dynamics of DOC degradation. This process has higher rates in the spring freshet that later in the summer, and the different pools of marine and riverine DOC have different degrees of bioavailabilty. I didn't see this addressed significantly"

According to this comment, we improved the first paragraph of the Perspective section. The text was modified as follows: "In addition, the model involves some limitations mostly due to the biogeochemical processing of RDOC, which is complex to translate into robust mechanistic equations as highly dependent on the availability of in-situ data in Arctic waters. For instance, the RDOC compartment is split in the model into

a labile and a non-labile fraction (see Le Fouest et al., 2015). This parameterization strongly constrains the removal of RDOC by bacterioplankton and therefore the RDOC concentrations simulated within surface waters. In natural waters, however, RDOC is made of a complex mixture of compounds that differ by their chemical composition and age (Mann et al., 2016) and so along the seasons (Wickland et al., 2012, Mann et al., 2012). The chemical nature of RDOC impacts its bioavailability estimated to average 6% to 46% of the total RDOC pool with marked disparities amongst the seasons and the rivers (Mann et al., 2012). Nevertheless, the general trend for the six major Arctic rivers (Kolyma, Yukon, Mackenzie, Ob, Yenisey and Lena) is a more labile RDOC pool in winter than in spring and summer (Wickland et al., 2012). Mann et al. (2012) report that, in the Kolyma River, the labile fraction is higher in spring (∼20%) than in summer (<10%) as the exported RDOC is younger during the freshet. Such a pattern is, however, not clearly evidenced in the Mackenzie River (e.g. Wickland et al., 2012). We suggest that a more realistic representation in the model of the nature of the organic matter entering the coastal waters including the riverine flux of both dissolved organic carbon and nitrogen along with an improved C:N stoichiometry for bacterioplankton uptake (see Le Fouest et al., 2015) might improve the RDOC concentrations simulated in surface AO waters."

3. "I didn't see this addressed significantly, including including the extent to which DOC is removed in the river delta or near-shore zone, and after it is accounted for in flux estimates, but before it reaches the open ocean where estimates can be made from satellite platforms."

Line 278 was also modified as follows: "In the model, the seasonal forcing of RDOC was based on RDOC measurements gathered hundreds kilometers upstream the rivers' mouths. This prevented any DOC enrichment of the Mackenzie River water as it flows through the delta (e.g. Emmerton et al., 2008) with, as a consequence, a likely underestimation of RDOC concentrations simulated in nearshore waters. Therefore, the quantification of the RDOC flux from the watersheds to the coastal AO poses

as another key issue to addressing its role in the biogeochemistry of shelf waters."

4. "It is also significant that much of the spring freshet flows over and under coastal sea ice from the Mackenzie River, but there is little inference about how that is accounted for."

In the model, the river flow and RDOC concentrations spread under coastal sea ice. By contrast, there were no RDOC data where sea ice above the sea surface was present in the remote sensing dataset. Therefore, only the grid points where both simulated an remotely sensed RDOC coincided were analyzed.

5. "Comparisons are made to primary production, and it is stated that DOC from rivers represents 10-19% of the carbon fixed by primary production in the Arctic Ocean as a whole and up to 34% of primary production in the coastal Beaufort Sea, but the labile nature of organic carbon that is formed by marine production is quite different from most of the organic carbon in RDOC. "

The reviewer's comment is relevant in the sense that most of RDOC is refractory to biological use while biogenic carbon formed by primary producers can flow more easily within the food webs when it does not sink out of the euphotic zone ($\sim$10%; Buesseler, 1998). However, the purpose of our sentence was primarily to scale a bulk comparison between these two main sources of organic carbon that fuel the upper water column, irrespective to their nature and fate. We hence modified the sentence as follows: "Irrespective to their distinct nature and fate, both the carbon contained in RDOC and that formed by primary producers can be considered as new carbon fueling annually the upper AO. For comparison, RDOC would amount $\sim$10% to 19% of the carbon formed by phytoplankton in the whole AO (Stein and Macdonald, 2004; Bélanger et al., 2013), a proportion that would reach $\sim$34% in the oligotrophic Beaufort Sea (S. Bélanger, pers. comm.)"

6. "It should be mentioned that the authors acknowledge some of these limitations in a general sense, including seasonal challenges to gathering satellite data, and the

complex nature of RDOC in the Perspectives section, although those limitations are not reflected in the abstract of the study, which reads more optimistically."

The limitations pointed out be the reviewer were developed in our detailed answers to comments 2 and 3. We also modified the beginning of the third paragraph of the Perspective section (line 288) to develop on some aspects of the remote sensing: "In our study, the RDOC concentrations remotely sensed in shelf waters provide the advantage of already integrating the effect of the watersheds processes such as mobilization, transformation and transport at the seasonal and synoptic time scales. However, we acknowledge that the temporal coverage of the remote sensing data is restricted to spring and summer and that we miss the cloudy and ice-dominated winter season, when RDOC is the most labile (e.g. Wickland et al., 2012) and likely subject to degradation within surface waters. In the Mackenzie River, the winter season is responsible for ∼25% of the annual load of labile RDOC (Wickland et al., 2012). Despite this limitation, and in regard to the model-satellite data comparison, the assimilation of remotely sensed RDOC data into Arctic models would still offer an interesting perspective as it might result in more realistic simulated fields of RDOC in spring-summer, when the river discharge and RDOC export is the highest."

"The manuscript could be improved by light editing by a Native English language writer."

The English will be improved.

"Data supporting the study are available on-line, but no metadata or "read-me file" explaining use of the on-line files is provided."

A read-me file will be provided with the data.

"I see no reason the manuscript couldn't be improved and accepted for publication, but I am skeptical of its potential for providing a more transformative understanding of dissolved organic carbon cycling in the Arctic."

To our knowledge, this study is the first to compare cutting-edge RDOC data derived

from remote sensing datasets and outputs from a predictive Arctic model. We show that the two approaches compare favorably in terms of RDOC concentrations and lateral fluxes and that they could be associated to overcome, at least partially, their own limitations. The study also attempts to shed light on the potential to further develop the two approaches to contribute for a better understanding of the RDOC dynamics and fate within AO waters in past and future decades, and so along with the increasing sampling effort done in the Arctic. To that respect, we think this study could be relevant for publication.

New cited references Buesseler K. O.: The decoupling of production and particulate export in the surface ocean, Global Biogeochem. Cycles, 12, 297–310, 1998.

Emmerton, C. A., Lesack, L. F. W., and Vincent, W. F.: Nutrient and organic matter patterns across the Mackenzie River, estuary and shelf during the seasonal recession of sea-ice, J. Marine Syst., 74, 741–755, doi:10.1016/j.jmarsys.2007.10.001, 2008.

---

## Author Comment (AC2) · 4 Oct 2017

We gratefully thank referee #2 for her/his constructive comments with respect to our manuscript. In order to improve the manuscript with respect to these comments, we amended the manuscript as suggested by the referee wherever it was possible. Note that, when needed, comments were merged together to bring more clarity in the answer:

1. "I see the manuscript strengths as showing the direction of travel for this type of

research, so as such I would like to see the later section about future directions to be strengthened, and more definitive suggestions provided."

Line 302, we added this text to introduce the last paragraph of the section Perspectives: "Improving the capability of Arctic models to resolve the fate and pathways of RDOC in the AO will require certain limitations to be unlocked. To this purpose, future model developments should lie on the always increasing observational effort realized by mean of field campaigns and new remote sensing techniques. As a prerequisite, we can reasonably encourage improvements of the riverine forcings to better encompass the seasonal to interannual variability of the terrigenous dissolved organic matter, both in qualitative and quantitative terms. We also suggest bacterioplankton dynamics to be better represented in biogeochemical models. In particular, the processes related to the competition for resources, because dissolved organic carbon and nitrogen of both allochtonous and autochtonous origin are likely to play an important role in bacterioplankton growth in coastal waters impacted by river plumes."

2. "Further, it was not always clear to me what was new, or re-analysis of previously published research." "Line 70 - so this is the same biogeochemical model results from Le Fouest 2015? Please make this explicit here. What about the remote sensing component, is that new or also from previous work?"

The study of Le Fouest et al. (2015) analyzed model outputs (primary and bacterioplankton production) obtained from a model run described as the "RIV run" in Le Fouest et al. (2015). The current study used other output data from the same model run "RIV run" but that were not analyzed yet. Those include RDOC concentration and ocean currents. The remote sensing data are very new and based on the new methods recently published in Matsuoka et al. (2017).

For more clarity, the sentence was reworded as follows: "To this end, sea surface RDOC concentrations obtained from a previous model run described in Le Fouest et al. (2015) and derived from remote sensing data were analyzed for the Canadian

Beaufort Sea."

Introduction "Line 27 - no need for thus."

"thus" was removed.

"Line 31 - no need for riverine as implicit in RDOC."

"riverine" was removed.

"Line 32 - with *the* potential for fueling"

"a potential" was replaced by "the potential".

"Line 35 - Awkward ending. Maybe consider something like: "Future studies could apply...the entire AO to quantify.."

The sentence was modified as follows: "Future studies could apply the combination of model and satellite data extended to the entire AO to quantify the expected changes in RDOC fluxes and their potential impact on AO biogeochemistry.". The sentence "This is left for future work" was removed.

"Line 39 - did you mean from the *six* great Arctic rivers in this paper?"

The sentence has been modified as follows: "The Arctic Ocean (AO) receives ∼10% of the global freshwater discharge (Opsahl et al., 1999 and references therein) of which the larger part (∼54-64%) originates from six main pan-Arctic rivers (Haine et al., 2015; Holmes et al., 2012; Aagaard and Carmack, 1989)."

"Line 40-41 - other factors contribute to the this intensification e.g. snow cover reduction, terrestrial productivity changes. Needs more detail here or suggest that increasing precipitation is one example." "Line 41 - grammar needs correcting"

The sentence was modified as follows: "Over the past 30 years, the Arctic freshwater cycle intensified as reflected by changes in snow cover (Bring et al., 2016), evapotranspiration from terrestrial vegetation (Bring et al., 2016), and precipitation (Vihma et al.,

2016). It resulted into an increase of the freshwater discharge from North American and Eurasian rivers by ∼2.6% and ∼3.1% per decade, respectively (Holmes et al., 2015)."

"Line 43 – contains half the soil *organic C stock"

"soil carbon stock" was replaced by "soil organic carbon stock".

"Line 45 - maybe worth mentioning that it is currently unclear though if aquatic OC concentrations will increase, and that some studies suggest that OC concentrations may reduce (see Abbott et al 2016 and Striegl et al. 2005 for example)." "Line 46 - not particularly suitable reference for the later part of the sentence re. changing OC concentration and composition. Suggest you use another and move the excellent Romanovsky one earlier in sentence."

The last two sentences were modified as follows: "With the warming of the lower atmosphere, the permafrost undergoes a substantial thawing (Romanovsky et al., 2010) likely to alter the organic carbon (OC) content and quality of inland waters. In the past decades, the flux of riverine dissolved organic carbon (RDOC) decreased in the Yukon River (40%; Striegl et al., 2005) while it increased at the Mackenzie River mouth (∼39%; Tank et al., 2016). These contrasting trends reinforce the idea that the direction of future trends of RDOC concentrations and fluxes from land to ocean remains very uncertain (Abbott et al., 2016)."

"Line 49 - these rivers flow all year round, so OM supply does not only occur after the ice breakup period."

The text was modified as follows to give a general sense to the sentence: "Coastal waters influenced by river plumes are hence exposed to changing conditions in terms of OC flux from land. They are generally supplied in riverine OC all year round with a maximal flux in spring-early summer when the river discharge reaches a seasonal maximum. ". With respect to other coastal areas, the Beaufort Sea system is quite

particular as the inner Mackenzie shelf (< 20 m depth) is bounded during winter by a thick ridged ice barrier grounded on the sea floor called stamukhi (Macdonald et al., 1995). The stamukhi retains the turbid river water within the inner shelf in winter. When sea ice breaks up and the freshet reaches its seasonal maximum in June, the turbid waters retained inshore spread farther within the coastal zone. This part is developed in lines 161 to 165.

"Line 50 - seasonal in twice."

"seasonal river flow" was replaced by "river flow".

"Line 57 - unusual to have a pers comm here as well as the Manizza paper. Recommend removing as adds little evidence."

The sentences were modified as follows: "The pan-Arctic flux of RDOC ($\sim$35-37.7 TgC yr-1; Holmes et al., 2012; Manizza et al., 2009; Raymond et al., 2007; Opsahl et al., 1999) is hence a significant pool of the carbon cycle. For comparison, it represents $\sim$10% to $\sim$19% of the carbon fixed by phytoplankton in the whole AO (Stein and Macdonald, 2004; Bélanger et al., 2013) and reaches up $\sim$34% of primary production in the oligotrophic Beaufort Sea (S. Bélanger, pers. comm.)."

"Line 61 - can this be written more clearly. Its an important point, so how is RDOC reducing C uptake by 10%? Or is it offsetting this?"

The sentence was modified as follows: "Furthermore, RDOC can modulate the air-sea fluxes of $CO_2$ at the pan-Arctic scale. The mineralization of RDOC produces dissolved inorganic carbon with, as a result, a decrease by 10% of the net oceanic $CO_2$ uptake in present climatic conditions (Manizza et al., 2011)."

Materials and methods "90 - more details on the satellite products used and their source would be useful here." "93 - unclear grammar here so not sure how you are coming up with this uncertainty value."

The paragraph was modified as follows: "Monthly composites of remotely

sensed RDOC concentrations are calculated as follows: Level 1A scene im­ages acquired from the MODerate-resolution Imaging Spectroradiometer (MODIS) aboard Aqua satellite were downloaded from the NASA ocean color website (https://oceandata.sci.gsfc.nasa.gov/MODIS-Aqua/L1/). Temporal data covered from June to September for the 2003-2013 period. After geometric correction, remote sens­ing reflectance, Rrs($\lambda$) data at 412, 443, 488, 531, 555, and 667 nm were obtained by applying atmospheric correction proposed by Wang and Shi (2009) with modifica­tions adapted to Arctic environments (Doxaran et al., 2015; Matsuoka et al., 2016). The light absorption coefficients of colored dissolved organic matter at 443 nm (aC­DOM(443)) were derived from the Rrs($\lambda$) data using the gsmA algorithm (Matsuoka et al., 2017) that optimizes the difference between satellite Rrs($\lambda$) and Rrs($\lambda$) calcu­lated using parameterization of absorption and backscattering coefficients for Arctic waters (Matsuoka et al., 2011, 2013). RDOC concentrations were estimated from the aCDOM(443) data using an empirical relationship between RDOC and aCDOM(443) established in the Southern Beaufort Sea (Matsuoka et al., 2013). Scene images of DOC concentrations were used to make monthly composite images at 1 km horizon­tal resolution. Errors of intercept, slope, and aCDOM(443) were propagated into the in-situ (empirical) DOC versus aCDOM(443) relationship. Mean uncertainty of DOC concentration estimates was hence determined to be 28% according to statistical anal­ysis(see Appendix A2 of Matsuoka et al., 2017)."

"97 - so are you including new model runs here or are they the same as subsequently published?"

The first two sentences were modified as follows: "We used sea surface RDOC con­centrations and ocean currents simulated by a pan-Arctic model run described in Le Fouest et al. (RIV run; 2015). The model data were extracted on the remote sensing geographical domain focused on the southern Beaufort Sea. We provide here a brief description of the physical-biogeochemical coupled model. The MITgcm (MIT general circulation model) ocean-sea ice model (Nguyen et al., 2011, 2009; Losch et al., 2010;

Condron et al., 2009) has a variable horizontal resolution of ∼18 km and covers the Arctic domain with open boundaries at 55°N on the Atlantic Ocean and Pacific Ocean sides."

"112 - please state how Raymond calculated this estimate."

The sentence was modified as follows: "The total annual load of RDOC in the model is 37.7 TgC yr-1. It is consistent with the values reported by Raymond et al. (36 TgC yr-1; 2007) and Holmes et al. (34 TgC yr-1; 2012) and obtained by load estimation models linking the RDOC concentrations to river discharge data. "

"119 - does Wickland really show this? I think she shows that between 12-18% of RDOC is available but that the average % is 15% in the Yukon river only. Please provide detail on assumptions."

The 15% value given in the manuscript was estimated using the yearly mean percentages of the total RDOC load considered as biodegradable DOC for six major Arctic rivers (Kolyma, Yukon, Mackenzie, Ob, Yenisey and Lena) given in Table 5 in Wickland et al. (2012).

The sentence was modified as follows: "We set to 15% the percentage of RDOC entering the model as usable by the bacterioplankton compartment. This value was estimated using the yearly mean percentages of the total RDOC load considered as biodegradable DOC for six major Arctic rivers given in Wickland et al. (2012)."

"136 - please reword this sentence for clarity."

The sentence was modified as follows: "Monthly fluxes of RDOC were calculated along two cross-shelf transects (see upper-middle panel in Fig. 1). The model estimates were computed as the product of the simulated sea surface current velocity with the simulated RDOC concentration. The remote sensing estimates were computed as the product of the simulated sea surface current velocity with the remotely sensed RDOC concentrations."

Results & Discussion "146 - you define an acronym for simulated RDOC (RDOCsim) in the methods but then don't use it in this section."

RDOC will be substituted by DOCt and, as such, RDOCsim will be removed.

"148 - quite speculative this. Are you suggesting that this may account for the differences and can you justify this with any estimates? Most would not consider ice-derived plankton terrestrially derived also, so please re-phrase." "156 – ok so here you say this is not likely to be the cause."

The sentence "Terrigenous DOC originating from both melted sea ice and permafrost erosion along the coastline were not taken into account in the model." was removed. The text was modified as follows to bring more clarity: "In the Beaufort and Chukchi seas, first year sea ice represents a carbon flux to the ocean of $2 \times 10^{-4}$ TgC yr-1 (Rachold et al., 2004). This flux is 4 orders of magnitude lower than the RDOC supply from the Mackenzie River specified as boundary conditions in the model (2.54 TgC yr-1). Similarly, DOC eroded from permafrost stored in the Canadian Arctic shores would account for only $\sim 0.5 \times 10^{-4}$ (Tanski et al., 2016) to $\sim 1.6 \times 10^{-4}$ TgC yr-1 (Ping et al., 2011, using a DOC:POC ratio of 1:900 as in Tanski et al., 2016). With regard to these flux values, terrigenous DOC originating from both melted sea ice and permafrost erosion along the coastline, not taken into account in the model, are hence not believed to explain the model-satellite discrepancies (Fig. 1). "

"150 & 154 - should this read 2 x 10? Please update."

"2 10-4 TgC yr-1" was replaced by "$2 \times 10^{-4}$ TgC yr-1". "0.5 10-4" and "1.6 10-4 TgC yr-1" were replaced by "$0.5 \times 10^{-4}$" and "$1.6 \times 10^{-4}$ TgC yr-1", respectively.

"157 (e.g. ??)"

The factors potentially involved to explain the model-satellite discrepancies are developed within the paragraph just after this sentence.

"162 - less than 20 m of depth/ distance?"

The sentence was modified as follows: "Second, the inner Mackenzie shelf (< 20 m depth) is bounded during winter by a thick ridged ice barrier grounded on the sea floor called stamukhi (Macdonald et al., 1995)."

"168 - Further offshore?"

The sentence was modified as follows: "Further offshore on the Mackenzie shelf, as delimited by the 300 m isobaths remotely sensed and simulated concentrations of RDOC were both within the range of values measured in spring (∼110-230 mmolC m-3; Osburn et al., 2009) and summer (∼60-100 mmolC m-3; Para et al., 2014)."

"183 - I'm not clear on how this works? RMSE shows that the model was more 'accurate' after the spring flush. Yet, the MEF index shows that model and observations were closest during and just after the flush? Can you explain the discrepancy here, or am I misunderstanding?" "184 - why does a positive MEF indicate this?" "195 - please re-word to make this sentence clearer."

A cross-verification of the metrics revealed an small error in the calculation of the geometric bias and RMSE shown in Table 1. It resulted into only a slight departure from the original values. We provide the corrected values in the new Table 1 below:

Table 1. Skill metrics of comparison computed based on the 2003-2011 monthly climatologies of RDOC. Metric June July August September Correlation coefficient 0.79 0.82 0.78 0.79 Unbiased RMSE (mmolC m-3) 41.4 29.4 26.0 29.3 Model efficiency 0.49 0.60 0.26 0.38

Geometric statistics using log-transformed data Model bias 1.24 1.07 1.32 1.21 RMSE 1.07 1.02 1.12 1.06

The text was modified as follows: "The size of the model-satellite discrepancies was given by the unbiased RMSE. Overall, the unbiased RMSE decreased from June (41.4 mmolC m-3) to September (29.3 mmolC m-3). This result suggested that the model accuracy increased from spring, i.e. during seasonal peak of river discharge in agreement

with Manizza et al. (2009) and Yang et al. (2015), to summer. The model capability for predicting RDOC relative to the average of the remote sensing counterparts was estimated by the model efficiency index (-∞<MEF≤1) (Nash and Sutcliffe, 1970).The MEF is a normalized statistic that relates the residual variance between the simulated and remotely sensed RDOC concentrations to the variance within the remotely sensed RDOC data. A MEF near zero means that the residual variance compares to the remotely sensed variance, i.e. and that the model predictions are as accurate as the mean of the satellite data. As the MEF increases towards a value of one, the residual variance becomes increasingly lower than the observed variance. The MEF was positive (0.26-0.60) for all months suggesting that RDOC concentrations simulated by the model were an acceptable predictor relative to RDOC concentrations derived from remote sensing, especially in June-July. Metrics based on log-transformed RDOC data were also computed to give a more even weight to all of the data and to limit the skewness towards the higher RDOC concentrations. For all months, the geometric RMSE was close to one and span between 1.02 and 1.12. It suggested that the model-satellite data dispersion was relatively small when the positive skewness was reduced. In June, the relatively high unbiased RMS could be partly due to high RDOC concentrations as suggested by relatively low geometric RMSE (1.07). Finally, the geometric bias informs on the direction of the model-satellite discrepancies. For all months, the geometric bias (1.07-1.32) was higher than one meaning that the model tended, on average, to overestimate the observations over the whole domain. The highest geometric bias was reported in August (1.32), when the river discharge was low, suggesting that RDOC removal was likely underestimated in the model in late summer. "

"General Text could benefit from editing for English grammar."

The English will be improved.

"References are not in alphabetical order in places e.g. Raymond ref higher up etc."

References will be sorted in alphabetical order.

"Is it appropriate to use RDOC as a term for the flux of C in the shelf region when it may be derived of a significant proportion of non riverine-derived OC?"

In the model, the RDOC compartment refers to DOC of riverine origin only.

With respect to remote sensing, the algorithm used in Matsuoka et al. (2017) was based on a valid and highly significant relationship between DOC and aCDOM(443) ($r2=0.97, p<0.0001$; Fig. 9a of Matsuoka et al., 2012). This type of relationship can only be observed in a river mouth. Because the algorithm was dependent on aCDOM(443) (aCDOM(443) versus salinity relationship: $r2=0.95, p<0.0001$, Fig. 5a of Matsuoka et al., 2012), the estimated relative fraction of terrestrial DOC retrieved would be $\sim0.92$, i.e. the product of $r2=0.95$ (from the aCDOM(443) versus salinity relationship) with $r2=0.97$ (from the DOC versus aCDOM(443) relationship). Note that a direct relationship between DOC and salinity ($r2=0.89, p<0.0001$; Fig. 8a of Matsuoka et al., 2012) confirmed that $\sim90\%$ of DOC observed in the river mouth was of terrestrial origin. So it can safely be argued that most ($\sim90\%$) of the DOC that was estimated by remote sensing was of terrestrial origin. This new section will be added in the text in section 2.1.

Nevertheless, we will replace the term RDOC by DOCt (for terrigenous DOC) for more accuracy.

New cited references

Aagaard, K. and Carmack, E. C.: The role of sea ice and other fresh water in the Arctic circulation, J. Geophys. Res., 94, doi:10.1029/JC094iC10p14485. Issn: 0148-0227, 1989.

Abbott, B. W., Jones, J. B., Schuur, E. A. G., Chapin, F. S., III, Bowden, W. B., Bret-Harte, M. S., et al.: Biomass offsets little or none of permafrost carbon release from soils, streams, and wildfire: an expert assessment, Environmental Discussion paper Research Letters, 11(3), 034014–14, doi:10.1088/1748-9326/11/3/034014, 2016.

Bring, A., Fedorova, I., Dibike, Y., Hinzman, L., Mård, J., Mernild, S. H., Prowse, T., Semenova, O., Stuefer, S. L., and Woo M.-K.: Arctic terrestrial hydrology: A synthesis of processes, regional effects, and research challenges, J. Geophys. Res. Biogeosci., 121, 621–649, doi:10.1002/2015JG003131, 2016.

Doxaran D., Devred, E. C., and Babin, M.: A 50 % increase in the mass of terrestrial particles delivered by the Mackenzie River into the Beaufort Sea (Canadian Arctic Ocean) over the last 10 years, Biogeosci., 12, doi:10.5194/bg-12-3551-2015, 3551-3565, 2015.

Haine, T. W. N., Curry, B., Gerdes, R., Hansen, E., Karcher, M., Lee, C., Rudels, B., Spreen, G., de Steur, L., Stewart, K. D., and Woodgate R.: Arctic freshwater export: Status, mechanisms, and prospects, Global and Planetary Change, 125, 13–35,doi:/10.1016/j.gloplacha.2014.11.013, 2015.

Matsuoka, A., Boss, E., Babin, M., Karp-Boss, L., Hafez, M., Chekalyuk, A., Proctor, C. W., Werdell, P. J., and Bricaud, A.: Pan-Arctic optical characteristics of colored dissolved organic matter: Tracing dissolved organic carbon in changing Arctic waters using satellite ocean color data, Remote Sens. Env., 200, 89–101, doi:10.1016/j.rse.2017.08.009, 2017.

Matsuoka, A., Babin, M., and Devred, E. C.: A new algorithm for discriminating water sources from space: a case study for the southern Beaufort Sea using MODIS ocean color and SMOS salinity data, Remote Sens. Env., 184, 124–138, http://dx.doi.org/10.1016/j.rse.2016.05.006, 2016.

Matsuoka, A., Hooker, S. B., Bricaud, A., Gentili, B., and Babin M.: Estimating absorption coefficients of colored dissolved organic matter (CDOM) using a semi-analytical algorithm for Southern Beaufort Sea waters: application to deriving concentrations of dissolved organic carbon from space, Biogeosci., 10, 917–927, doi:10.5194/bg-10-917-2013, 2013.

Matsuoka, A., Bricaud, A., Benner, R., Para, J., Sempéré, R., Prieur, L., Bélanger, S., and Babin, M.: Tracing the transport of colored dissolved organic matter in water masses of the Southern Beaufort Sea: relationship with hydrographic characteristics, Biogeosci., 9, 925-940, doi:10.5194/bg-9-925-2012, 2012.

Matsuoka, A., Hill, V., Huot, Y., Babin, M., and Bricaud, A.: Seasonal variability in the light absorption properties of Western Arctic waters: parameterization of the individual components of absorption for ocean color applications, J. Geophys. Res., 116, C02007, doi:10.1029/2009JC005594, 2011.

Opsahl, S., Benner, R., and Amon, R. M. W.: Major flux of terrigenous dissolved organic matter through the Arctic Ocean, Limnol. Oceanogr., 44, 2017–2023, 1999.

Striegl, R. G., Aiken, G. R., Dornblaser, M. M., Raymond, P. A., and Wickland, K. P.: A decrease in discharge-normalized DOC export by the Yukon River during summer through autumn, Geophysical Research Letters, 32(21), L21413, doi:10.1029/2005GL024413, 2005.

Vihma, T., Screen, J., Tjernström, M., Newton, B., Zhang, X., Popova, V., Deser, C., Holland, M., and Prowse, T.: The atmospheric role in the Arctic water cycle: A review on processes, past and future changes, and their impacts, J. Geophys. Res. Biogeosci., 121, 586–620, doi:10.1002/2015JG003132, 2016.

Wang, M. and Shi, W.: The NIR-SWIR combined atmospheric correction approach for MODIS ocean color data processing, Opt. Express, 15, 15722–15722, 2007.

---

## Author Response (AR1)

Dr. Vincent Le Fouest, Associate Professor
LIttoral ENvironnement et Sociétés (LIENSs) - UMR 7266
Université de la Rochelle, Bâtiment ILE
2, rue Olympe de Gouges
17000 La Rochelle
France
Email: vincent.le_fouest@univ-lr.fr

La Rochelle, November 8, 2017

Object: Revision of the manuscript bg-2017-286

Dear Editor,

Please find attached a revised version of the manuscript entitled "Towards an assessment of riverine dissolved organic carbon in surface waters of the Western Arctic Ocean based on remote sensing and biogeochemical modeling" by V. Le Fouest, A. Matsuoka, M. Manizza, M. Shernetsky, B. Tremblay, and M. Babin. Based on your recommendations about the manuscript # bg-2017-286, we thank you to allow us providing a revised version of the manuscript which takes into account all the reviewers' comments. Following your request, we provide below a point-by-point response to the reviewers and a list of all relevant changes made in the manuscript. The changes corresponding to the major comments of both reviewers are coloured in red in the revised version.

Yours sincerely,

Dr. Vincent Le Fouest

We gratefully thank **referee #1** for her/his constructive comments with respect to our manuscript. In order to improve the manuscript with respect to these comments, we amended the manuscript as suggested by the referee wherever it was possible. Note that, when needed, comments were merged together to bring more clarity in the answer:

**1. "Satellite remote sensing obviously can convey information on the directional flow of river plumes carrying DOC, but depth penetration from satellite platforms is modest, so without field sampling, comparison of one set of estimates with another produced by biogeochemical modeling seems like a limited and incomplete outcome."**

We agree with that comment in the sense that numerical modeling and remote sensing are not exhaustive approaches. Both are fully dependent from field measurements as their setup (e.g. forcings and differential equations for the model, algorithms for remote sensing) necessarily requires large in-situ databases. In our study, the model was constrained by riverine DOC observations at the boundaries of its numerical domain (see Manizza et al., 2009) while the semi-empirical remote sensing algorithm we used was developed based on field measurements (see Matsuoka et al., 2017). As explicitly mentioned in the manuscript, the modeling and remote sensing approaches combined together provide a relevant insight on the terrigenous DOC (tDOC) dynamics over a wide spatial and temporal scale, but limited to the surface coastal waters where tDOC concentrations are the highest.

In order to account for the referee's comment, we modified the end of the abstract (lines 33-36) as follows: "The combination of model and satellite data provide promising results to extend this work to the entire AO so as to quantify, in conjunction with in-situ data, the expected changes in tDOC fluxes and their potential impact on the AO biogeochemistry at basin scale."

**2. "Moreover, many of the important areas of concern in the context of climate change revolve around the dynamics of DOC degradation. This process has higher rates in the spring freshet that later in the summer, and the different pools of marine and riverine DOC have different degrees of bioavailabilty. I didn't see this addressed significantly"**

According to this comment, we improved the first paragraph of the Perspective section. The text was modified as follows (lines 298-316): "In addition, the model involves some limitations mostly due to the biogeochemical processing of tDOC, which is complex to translate into robust mechanistic equations as highly dependent on the availability of in-situ data in Arctic waters. For instance, the riverine tDOC compartment is split in the model into a labile and a non-labile fraction (see Le Fouest et al., 2015). This parameterization strongly constrains the removal of tDOC by bacterioplankton and therefore the tDOC concentrations simulated within surface waters. In natural waters, however, tDOC is made of a complex mixture of compounds that differ by their chemical composition and age (Mann et al., 2016) and so along the seasons (Wickland et al., 2012, Mann et al., 2012). The chemical nature of tDOC impacts its bioavailability estimated to average 6 % to 46 % of the total tDOC pool with marked disparities amongst the seasons and the rivers (Mann et al., 2012). Nevertheless, the general trend for the six major Arctic rivers (Kolyma, Yukon, Mackenzie, Ob, Yenisey and Lena) is a more labile tDOC pool in winter than in spring and summer (Wickland et al., 2012). In the Kolyma River, Mann et al. (2012) report a higher labile fraction in spring (~20 %) than in summer (<10 %) as the exported tDOC is younger during the freshet. Such a pattern is, however, not clearly evidenced in the Mackenzie River (e.g. Wickland et al., 2012). We suggest that a more realistic representation in the model of the nature of the organic matter entering the coastal waters including the riverine flux of both dissolved organic carbon and nitrogen along with an improved C:N stoichiometry for bacterioplankton uptake (see Le Fouest et al., 2015) might improve the tDOC concentrations simulated in surface AO waters."

**3. "I didn't see this addressed significantly, including including the extent to which DOC is removed in the river delta or near-shore zone, and after it is accounted for in flux estimates, but before it reaches the open ocean where estimates can be made from satellite platforms."**

The text was also modified as follows (lines 317-322): "In the model, the seasonal forcing of tDOC was based on DOC measurements gathered hundreds kilometers upstream the rivers' mouths. This prevented any DOC enrichment of the Mackenzie River water as it flows through the delta (see Emmerton et al., 2008) with, as a consequence, a likely underestimation of tDOC concentrations simulated in nearshore waters. Therefore, the quantification of the tDOC flux from the watersheds to the coastal AO poses as another key issue to addressing the role of tDOC in the biogeochemistry of shelf waters."

**4. "It is also significant that much of the spring freshet flows over and under coastal sea ice from the Mackenzie River, but there is little inference about how that is accounted for."**

In the model, the river flow and tDOC concentrations spread under coastal sea ice. By contrast, there were no tDOC data where sea ice above the sea surface was present in the remote sensing dataset. Therefore, only the grid points where both simulated an remotely sensed tDOC coincided were analyzed.

**5. "Comparisons are made to primary production, and it is stated that DOC from rivers represents 10-19% of the carbon fixed by primary production in the Arctic Ocean as a whole and up to 34% of primary production in the coastal Beaufort Sea, but the labile nature of organic carbon that is formed by marine production is quite different from most of the organic carbon in RDOC. "**

The reviewer's comment is relevant in the sense that most of tDOC is refractory to biological use while biogenic carbon formed by primary producers can flow more easily within the food webs when it does not sink out of the euphotic zone (~10%; Buesseler, 1998). However, the purpose of our sentence was primarily to scale a bulk comparison between these two main sources of organic carbon that fuel the upper water column, irrespective to their nature and fate. We hence modified the sentence as follows (lines 59-64): "As the organic carbon formed by phytoplankton, terrigenous DOC (tDOC) can be considered as new carbon fueling annually the upper AO. To that respect, and regardless of its distinct nature and fate, the flux of riverine DOC would be equivalent to 10-19 % of the AO primary production (Stein and Macdonald, 2004; Bélanger et al., 2013). In the oligotrophic Beaufort Sea, this proportion would reach ~34 % (S. Bélanger, pers. comm.)."

**6. "It should be mentioned that the authors acknowledge some of these limitations in a general sense, including seasonal challenges to gathering satellite data, and the complex nature of RDOC in the Perspectives section, although those limitations are not reflected in the abstract of the study, which reads more optimistically."**

The limitations pointed out be the reviewer were developed in our detailed answers to comments 2 and 3. We also modified the beginning of the third paragraph of the Perspective section (lines 331-340) to develop on some aspects of the remote sensing: "In our study, the remotely sensed tDOC concentrations retrieved in shelf waters provide the advantage of already integrating the effect of the watersheds processes such as mobilization, transformation and transport at the seasonal and synoptic time scales. However, we acknowledge that the temporal coverage of the remote sensing data is restricted to spring and summer. Because of clouds and sea ice, we miss the winter season when tDOC is the most labile (e.g. Wickland et al., 2012) and likely subject to remineralization. In the Mackenzie River, about 25 % of the annual load of labile tDOC occurs during winter (Wickland et al., 2012). Despite this limitation, and in regard to the model-satellite data comparison, the assimilation of remotely sensed tDOC data into Arctic models could still offer an interesting perspective as it might result in more realistic simulated fields of tDOC in spring and summer when the river discharge and tDOC export is the highest."

**7. "The manuscript could be improved by light editing by a Native English language writer."**

The English has been improved.

**8. "Data supporting the study are available on-line, but no metadata or "read-me file" explaining use of the on-line files is provided."**

A read-me file is now provided with the data in the archive.

**9. "I see no reason the manuscript couldn't be improved and accepted for publication, but I am skeptical of its potential for providing a more transformative understanding of dissolved organic carbon cycling in the Arctic."**

To our knowledge, this study is the first to compare cutting-edge tDOC data derived from remote sensing datasets and outputs from a predictive Arctic model. We show that the two approaches compare favorably in terms of tDOC concentrations and lateral fluxes and that they could be associated to overcome, at least partially, their own limitations. The study also attempts to shed light on the potential to further develop the two approaches to contribute for a better understanding of the tDOC dynamics and fate within AO waters in past and future decades, and so along with the increasing sampling effort done in the Arctic. To that respect, we think this study could be relevant for publication.

**New cited references**
Buesseler K. O.: The decoupling of production and particulate export in the surface ocean, Global Biogeochem. Cycles, 12, 297–310, 1998.

Emmerton, C. A., Lesack, L. F. W., and Vincent, W. F.: Nutrient and organic matter patterns across the Mackenzie River, estuary and shelf during the seasonal recession of sea-ice, J. Marine Syst., 74, 741–755, doi:10.1016/j.jmarsys.2007.10.001, 2008.

We gratefully thank **referee #2** for her/his constructive comments with respect to our manuscript. In order to improve the manuscript with respect to these comments, we amended the manuscript as suggested by the referee wherever it was possible. Note that, when needed, comments were merged together to bring more clarity in the answer:

**"I see the manuscript strengths as showing the direction of travel for this type of research, so as such I would like to see the later section about future directions to be strengthened, and more definitive suggestions provided."**

We added this text to introduce the last paragraph of the Perspectives section (lines 349-359): "Improving the capability of Arctic models to resolve the fate and pathways of tDOC in the AO will require certain limitations to be unlocked. To this purpose, future model developments should lie on the always increasing observational effort realized by mean of field campaigns and new remote sensing techniques. As a prerequisite, we can reasonably encourage improvements of the riverine forcings to better encompass the seasonal to interannual variability of the terrigenous dissolved organic matter exported to the coastal AO, both in qualitative and quantitative terms. We also suggest bacterioplankton dynamics to be better represented in biogeochemical models. In particular, the processes related to the competition for resources as dissolved organic carbon and nitrogen of both allochtonous and autochtonous origin are likely to play an important role in mediating bacterioplankton growth and tDOC remineralization in Arctic coastal waters impacted by river plumes."

**"Further, it was not always clear to me what was new, or re-analysis of previously published research."**
**"Line 70 - so this is the same biogeochemical model results from Le Fouest 2015? Please make this explicit here. What about the remote sensing component, is that new or also from previous work?"**

The study of Le Fouest et al. (2015) analyzed model outputs (primary and bacterioplankton production) obtained from a model run described as the "RIV run" in Le Fouest et al. (2015). The current study used other output data from the same model run "RIV run" but that were not analyzed yet. Those include terrigenous DOC (tDOC) concentrations and ocean currents. The remote sensing data are very new and based on the new methods recently published in Matsuoka et al. (2017).

For more clarity, the sentence was reworded as follows (lines 79-81): "To this end, riverine DOC concentrations at the sea surface obtained from a previous model run described in Le Fouest et al. (2015) and tDOC concentrations derived from remote sensing data were analyzed for the Canadian Beaufort Sea."

**Introduction**
**"Line 27 - no need for thus."**

"thus" was removed.

**"Line 31 - no need for riverine as implicit in RDOC."**

"riverine" was removed.

**"Line 32 - with *the* potential for fueling"**

"a potential" was replaced by "the potential".

**"Line 35 - Awkward ending.  Maybe consider something like:  "Future studies could apply...the entire AO to quantify.."**

The sentence was modified as follows (lines 33-36): "The combination of model and satellite data provide promising results to extend this work to the entire AO so as to quantify, in conjunction with in-situ data, the expected changes in tDOC fluxes and their potential impact on the AO biogeochemistry at basin scale.". The sentence "This is left for future work" was removed.

**"Line 39 - did you mean from the *six* great Arctic rivers in this paper?"**

The sentence has been modified as follows (lines 39-41): "The Arctic Ocean (AO) receives ~10% of the global freshwater discharge (Opsahl et al., 1999 and references therein) of which the larger part (~54-64%) originates from six main pan-Arctic rivers (Haine et al., 2015; Holmes et al., 2012; Aagaard and Carmack, 1989)."

**"Line 40-41 - other factors contribute to the this intensification e.g. snow cover reduction, terrestrial productivity changes. Needs more detail here or suggest that increasing precipitation is one example."**
**"Line 41 - grammar needs correcting"**

The sentence was modified as follows (lines 41-45): "Over the past 30 years, the Arctic freshwater cycle intensified as reflected by changes in snow cover (Bring et al., 2016), evapotranspiration from terrestrial vegetation (Bring et al., 2016), and precipitation (Vihma et al., 2016). It resulted into an increase of the freshwater discharge from North American and Eurasian rivers by ~2.6 % and ~3.1 % per decade, respectively (Holmes et al., 2015)."

**"Line 43 – contains half the soil *organic C stock"**

"soil carbon stock" was replaced by "soil organic carbon stock".

**"Line 45 - maybe worth mentioning that it is currently unclear though if aquatic OC concentrations will increase, and that some studies suggest that OC concentrations may reduce (see Abbott et al 2016 and Striegl et al.  2005 for example)."**
**"Line 46 - not particularly suitable reference for the later part of the sentence re.  changing OC concentration and composition. Suggest you use another and move the excellent Romanovsky one earlier in sentence."**

The last two sentences were modified as follows (lines 47-52): "With the warming of the lower atmosphere, the permafrost undergoes a substantial thawing (Romanovsky et al., 2010) likely to alter the organic carbon content and quality of inland waters. In the past decades, the flux of dissolved organic carbon (DOC) decreased in the Yukon River (40 %; Striegl et al., 2005) while it increased at the Mackenzie River mouth (~39 %; Tank et al., 2016). These contrasting responses to climate change suggest that the direction of future trends of DOC concentrations and fluxes to the AO are very uncertain (Abbott et al., 2016)."

**"Line 49 - these rivers flow all year round, so OM supply does not only occur after the ice breakup period."**

The text was modified as follows to give a general sense to the sentence (lines 53-55): "Coastal waters are supplied in riverine organic carbon all year round with a maximal flux in spring-early summer when the freshwater discharge reaches a seasonal maximum.". With respect to other coastal areas, the Beaufort Sea system is quite particular as the inner Mackenzie shelf (< 20 m depth) is bounded during winter by a thick ridged ice barrier grounded on the sea floor called stamukhi (Macdonald et al., 1995). The stamukhi retains the turbid river water within the inner shelf in winter. When sea ice breaks up and the freshet reaches its seasonal maximum in June, the turbid waters retained inshore spread farther within the coastal zone. This latter part is developed from line 188 to line 192 in section 3.1.

**"Line 57 - unusual to have a pers comm here as well as the Manizza paper. Recommend removing as adds little evidence."**

The sentences were modified as follows (lines 58-64): "The pan-Arctic flux of riverine DOC to the AO is estimated to 33-37.7 TgC yr-1 (Holmes et al., 2012; Manizza et al., 2009; McGuire et al., 2009; Raymond et al., 2007). As the organic carbon formed by phytoplankton, terrigenous DOC (tDOC) can be considered as new carbon fueling annually the upper AO. To that respect, and regardless of its distinct nature and fate, the flux of riverine DOC would be equivalent to 10-19 % of the AO primary production (Stein and Macdonald, 2004; Bélanger et al., 2013). In the oligotrophic Beaufort Sea, this proportion would reach ~34 % (S. Bélanger, pers. comm.)."

**"Line 61 - can this be written more clearly. Its an important point, so how is RDOC reducing C uptake by 10%? Or is it offsetting this?"**

The text was modified as follows (lines 67-70): "Riverine DOC can also modulate the air-sea fluxes of $CO_2$. In present climatic conditions, Manizza et al. (2011) suggest that the mineralization of riverine DOC into dissolved inorganic carbon would induce a 10 % decrease of the net oceanic $CO_2$ uptake at the pan-Arctic scale."

**Materials and methods**
**"90 - more details on the satellite products used and their source would be useful here."**
**"93 - unclear grammar here so not sure how you are coming up with this uncertainty value."**

The section 2.1 was modified as follows (lines 98-115): "Level 1A scene images acquired from the MODerate-resolution Imaging Spectroradiometer (MODIS) aboard Aqua satellite were downloaded from the NASA ocean color website (https://oceandata.sci.gsfc.nasa.gov/MODIS-Aqua/L1/). After geometric correction, remote sensing reflectance, Rrs(λ) data at 412, 443, 488, 531, 555, and 667 nm were obtained by applying atmospheric correction proposed by Wang and Shi (2009) with modifications adapted to Arctic environments (Doxaran et al., 2015; Matsuoka et al., 2016). The light absorption coefficients of colored dissolved organic matter at 443 nm ($a_{CDOM}(443)$) were derived from the Rrs(λ) data using the gsmA algorithm (Matsuoka et al., 2017) that optimizes the difference between satellite Rrs(λ) and Rrs(λ) calculated using parameterization of absorption and backscattering coefficients for Arctic waters (Matsuoka et al., 2011, 2013). tDOC concentrations were estimated from the $a_{CDOM}(443)$ data using an empirical relationship between DOC and $a_{CDOM}(443)$ established in the Southern Beaufort Sea (Matsuoka et al., 2013). Since DOC concentrations estimated using ocean color data are based on a highly significant DOC versus $a_{CDOM}(443)$ relationship ($R^2$ = 0.97; Matsuoka et al., 2012), the DOC is considered to be of terrestrial origin. Errors of intercept, slope, and $a_{CDOM}(443)$ were propagated into the in-situ (empirical) DOC versus $a_{CDOM}(443)$ relationship. It resulted into a mean uncertainty of the tDOC concentration estimates of 28 % (see Appendix A2 of Matsuoka et al., 2017). Scene images of tDOC concentrations were used to make monthly composite images at 1 km horizontal resolution."

**"97 - so are you including new model runs here or are they the same as subsequently published?"**

The first two sentences were modified as follows (lines 118-122): "We used sea surface tDOC concentrations and ocean currents simulated over 2003-2011 by a previous pan-Arctic model run ("RIV run") whose setup is fully detailed in Le Fouest et al. (2015). The pan-Arctic model data were extracted on the remote sensing geographical domain focused on the southern Beaufort Sea. We provide here a brief description of the physical-biogeochemical coupled model used to generate the "RIV run"."

**"112 - please state how Raymond calculated this estimate."**

The sentence was modified as follows (lines 134-137): "The total annual load of tDOC in the model is 37.7 TgC yr$^{-1}$. It is consistent with previous values reported in Raymond et al. (36 TgC yr$^{-1}$; 2007) and Holmes et al. (34 TgC yr$^{-1}$; 2012) and obtained by using load estimation models linking riverine DOC concentrations to river discharge data."

**"119 - does Wickland really show this?  I think she shows that between 12-18% of RDOC is available but that the average % is 15% in the Yukon river only. Please provide detail on assumptions."**

The 15% value given in the manuscript was estimated using the yearly mean percentages of the total RDOC load considered as biodegradable DOC for six major Arctic rivers (Kolyma, Yukon, Mackenzie, Ob, Yenisey and Lena) given in Table 5 in Wickland et al. (2012).

The sentence was modified as follows (lines 143-146): "We set to 15 % the percentage of tDOC entering the model as usable by the bacterioplankton compartment. This value was estimated based on the mean yearly percentages of the total load of riverine DOC considered as biodegradable DOC for six major Arctic rivers given in Wickland et al. (2012)."

**"136 - please reword this sentence for clarity."**

The sentence was modified as follows (lines 160-164): "Monthly fluxes of tDOC were calculated and summed along two cross-shelf transects (see upper-middle panel in Fig. 1). At each grid cell, the model flux estimate was computed as the product of the simulated sea surface current velocity with the simulated tDOC concentration. The remote sensing flux estimate was computed as the product of the simulated sea surface current velocity with the remotely sensed tDOC concentration."

**Results & Discussion**
**"146 - you define an acronym for simulated RDOC (RDOCsim) in the methods but then don't use it in this section."**

RDOC was substituted by tDOC and, as such, RDOCsim has been be removed.

**"148 - quite speculative this. Are you suggesting that this may account for the differences and can you justify this with any estimates? Most would not consider ice-derived plankton terrestrially derived also, so please re-phrase."**
**"156 – ok so here you say this is not likely to be the cause."**

The sentence "Terrigenous DOC originating from both melted sea ice and permafrost erosion along the coastline were not taken into account in the model." was removed.
The text was modified as follows to bring more clarity (lines 175-183): "In the Beaufort and Chukchi seas, first year sea ice represents a carbon flux to the ocean of $2 \times 10^{-4}$ TgC yr$^{-1}$ (Rachold et al., 2004). This flux is 4 orders of magnitude lower than the tDOC supply from the Mackenzie River specified as boundary conditions in the model (2.54 TgC yr$^{-1}$). Similarly, tDOC eroded from permafrost stored in the North American shores would account for only ~0.5-1.6 $\times 10^{-4}$ TgC yr$^{-1}$ (Tanski et al., 2016; Ping et al., 2011, using a DOC:POC ratio of 1:900 as in Tanski et al., 2016) to ~$2 \times 10^{-3}$ TgC yr$^{-1}$ (McGuire et al., 2009). With regard to these flux values, tDOC originating from both melted sea ice and eroded permafrost, not taken into account in the model, are hence not believed to explain the model-satellite discrepancies (Fig. 1). "

**"150 & 154 - should this read 2 x 10? Please update."**

Line 176, "2 10$^{-4}$ TgC yr$^{-1}$" was replaced by "2 × 10$^{-4}$ TgC yr$^{-1}$" and so on when necessary.

**"157 (e.g. ??)"**

The factors potentially involved to explain the model-satellite discrepancies are developed within the paragraph just after this sentence.

**"162 - less than 20 m of depth/ distance?"**

The sentence was modified as follows (line 188-190): "Second, the inner Mackenzie shelf (< 20 m depth) is bounded during winter by a thick ridged ice barrier grounded on the sea floor called stamukhi (Macdonald et al., 1995)."

**"168 - Further offshore?"**

The sentence was modified as follows (lines 195-198): "Further offshore on the Mackenzie shelf, as delimited by the 300 m isobaths both remotely sensed and simulated concentrations of tDOC were within the range of values measured in spring (~110-230 mmolC m$^{-3}$; Osburn et al., 2009) and summer (~60-100 mmolC m$^{-3}$; Para et al., 2014). "

**"183 - I'm not clear on how this works? RMSE shows that the model was more 'accurate' after the spring flush. Yet, the MEF index shows that model and observations were closest during and just after the flush? Can you explain the discrepancy here, or am I misunderstanding?"**
**"184 - why does a positive MEF indicate this?"**
**"195 - please re-word to make this sentence clearer."**

A cross-verification of the metrics revealed a small error in the calculation of the geometric bias and RMSE shown in Table 1. It resulted into only a slight departure from the original values. We provide the corrected values in the new Table 1 below:

**Table 1.** Skill metrics of comparison computed based on the 2003-2011 monthly climatologies of tDOC.

| Metric | June | July | August | September |
|---|---|---|---|---|
| Correlation coefficient | 0.79 | 0.82 | 0.78 | 0.79 |
| Unbiased RMSE (mmolC m$^{-3}$) | 41.4 | 29.4 | 26.0 | 29.3 |
| Model efficiency | 0.49 | 0.60 | 0.26 | 0.38 |
| | | | | |
| Geometric statistics using log-transformed data | | | | |
| Model bias | 1.24 | 1.07 | 1.32 | 1.21 |
| RMSE | 1.07 | 1.02 | 1.12 | 1.06 |

The text was modified as follows (lines 205-227): "The size of the model-satellite discrepancies was given by the unbiased RMSE. Overall, the unbiased RMSE decreased from June (41.4 mmolC m$^{-3}$) to September (29.3 mmolC m$^{-3}$). This result suggested that the model accuracy increased from spring to summer. The model capability for predicting tDOC relative to the average of the remote sensing counterparts was estimated by the model efficiency index (-∞<MEF≤1) (Nash and Sutcliffe, 1970). The MEF is a normalized statistic that relates the residual variance between the simulated and remotely sensed tDOC concentrations to the variance within the remotely sensed tDOC data (see Eq. 3). A MEF value near zero means that the residual variance compares to the remotely sensed variance, i.e. that the model predictions are as accurate as the mean of the satellite data. As the MEF increases towards a value of one, the residual variance becomes increasingly lower than the observed variance. For all months, the MEF was positive (0.26-0.60) suggesting that tDOC concentrations simulated by the model were an acceptable predictor relative to tDOC concentrations derived from remote sensing, especially in June-July. In order to give a more even weight to all of the data and to limit the skewness towards the higher tDOC concentrations, metrics based on log-transformed tDOC data were also computed. For all months, the geometric RMSE was close to one and span between 1.02 and 1.12. It suggested that the model-satellite data dispersion was relatively small when the positive skewness was reduced. In June, the relatively high unbiased RMSE could be partly due to high tDOC concentrations as suggested by the relatively low geometric RMSE (1.07). Finally, the computed geometric bias informs on the direction of the model-satellite discrepancies. For all months, the geometric bias (1.07-1.32) was higher than one meaning that the model tended, on average, to overestimate the observations over the whole domain. The highest geometric bias was reported in August (1.32), when the river discharge was low, suggesting that tDOC removal was likely underestimated in the model in late summer."

**"General Text could benefit from editing for English grammar."**

The English has been improved.

**"References are not in alphabetical order in places e.g. Raymond ref higher up etc."**

References have been sorted in alphabetical order.

**"Is it appropriate to use RDOC as a term for the flux of C in the shelf region when it may be derived of a significant proportion of non riverine-derived OC?"**

In the model, the RDOC compartment refers to DOC of riverine origin only.

With respect to remote sensing, the algorithm used in Matsuoka et al. (2017) was based on a valid and highly significant relationship between DOC and $a_{CDOM}(443)$ ($R^2 = 0.97$, $p < 0.0001$; Fig. 9a of Matsuoka et al., 2012). This type of relationship can only be observed in a river mouth. Because the algorithm was dependent on $a_{CDOM}(443)$ ($a_{CDOM}(443)$ *versus* salinity relationship: $R^2 = 0.95$, $p < 0.0001$, Fig. 5a of Matsuoka et al., 2012), the estimated relative fraction of terrestrial DOC retrieved would be ~0.92, i.e. the product of $R^2 = 0.95$ (from the $a_{CDOM}(443)$ *versus* salinity relationship) with $R^2 = 0.97$ (from the DOC *versus* $a_{CDOM}(443)$ relationship). Note that a direct relationship between DOC and salinity ($R^2 = 0.89$, $p < 0.0001$; Fig. 8a of Matsuoka et al., 2012) confirmed that ~90 % of DOC observed in the river mouth was of terrestrial origin. So it can safely be argued that most (~90 %) of the DOC that was estimated by remote sensing was of terrestrial origin. This new section has been added in the text in section 2.1.

We have replaced the term RDOC by tDOC (for terrigenous DOC) for more accuracy.

[revised manuscript text omitted]

---

## Author Response (AR3)

Dr. Vincent Le Fouest, Associate Professor
LIttoral ENvironnement et Sociétés (LIENSs) - UMR 7266
Université de la Rochelle, Bâtiment ILE
2, rue Olympe de Gouges
17000 La Rochelle
France
Email: vincent.le_fouest@univ-lr.fr

La Rochelle, December 21, 2017

Object: Revision of the manuscript bg-2017-286

Dear Editor,

Please find attached a second revised version of the manuscript entitled "Towards an assessment of riverine dissolved organic carbon in surface waters of the Western Arctic Ocean based on remote sensing and biogeochemical modeling" by V. Le Fouest, A. Matsuoka, M. Manizza, M. Shernetsky, B. Tremblay, and M. Babin. Based on your recommendations about the manuscript # bg-2017-286, we thank you to allow us providing a second revised version of the manuscript which takes into account all the reviewers' comments. Following your request, we provide below a point-by-point response to the reviewers and a list of all relevant changes made in the manuscript. The changes corresponding to the major comments of both reviewers are coloured in red in the revised version.

Yours sincerely,

Dr. Vincent Le Fouest

We gratefully thank **referee #1** for her/his new constructive comments with respect to our manuscript. In order to improve the manuscript with respect to these comments, we amended the manuscript as suggested by the referee wherever it was possible. Note that, when needed, comments were merged together to bring more clarity in the answer:

**Line 40. Add a comma prior to "of which"**
Done

**Line 44. Replace "into" with "in"**
Done

**Line 58. Add "be" prior to 33-37.7 TgC**
Done

**Line 60. Delete "as" before new carbon**
Done

**Line 61. Start the sentence with "In" rather than "To"**
Done

**Line 62. Delete "the" prior to AO primary productivity.**
Done

**Line 70. Change in East Siberian shelves to on East Siberian shelves**
Done

**Line 71. Delete "the" prior to sea surface**
Done

**Line 84. Add "in order" prior to "to assess"**
Done

**Line 99. Add "the" prior to Aqua satellite**
Done

**Line 101. Add "the" at the end of this line and prior to atmospheric correction.**
Done

**Line 114. Not clear to me what a scene image is**
A scene image is a snapshot taken by the satellite sensor. We choose not to change this term because it is specific to remote sensing.

**Line 220. "range" would be better here than "span"**
Done

**Line 223. Add "with respect to" prior to "the direction" and remove "on" prior to "the direction"**
Done

**Line 234. "June and August were very close months" is awkward phrasing.**

We modified the sentence as follows: "June and August showed similar values of correlation, RMSE, and normalized standard deviation despite distinct seasonal patterns of river discharge (high and low, respectively). By contrast, September showed the highest model-satellite data dispersion."

**Line 298-302. This is a very lengthy and complex sentence that should be broken up, for example after tDOC on line 299. Start a new sentence here that states that it is difficult to estimate biogeochemical processing of tDOC due to limited field data or something equivalent.**
We modified the text as follows: "In addition, the model involves some limitations mostly due to the biogeochemical processing of tDOC. The tDOC transformation is complex to translate into robust mechanistic equations as highly dependent on the availability of in-situ data in Arctic waters."

**Line 306. Add a comma and "which is" at the end of the line after bioavailability.**
Done

**Line 312. replace "evidenced" with "present"**
Done

**Line 313-316. Break this lengthy and complex sentence up into smaller parts.**
We modified this sentence as follows: "We suggest that a more realistic representation in the model of the nature of the organic matter entering the coastal waters might improve the tDOC concentrations simulated in surface AO waters. It could include, for instance, the riverine flux of both dissolved organic carbon and nitrogen along with an improved C:N stoichiometry for bacterioplankton uptake (see Le Fouest et al., 2015)"

**Line 318. "precludes" is better here than "prevented"**
Done

**Line 350-354. These two sentences are very awkwardly phrased and the second sentence is a run-on construction.**
We modified the text as follows: "To this purpose, future model developments must lie on the always increasing observational effort realized by mean of field campaigns and new remote sensing techniques. Observations must be used to improve the riverine forcings in order to better encompass the seasonal to interannual variability of the terrigenous dissolved organic matter exported to the coastal AO."

**Line 355. Rather that state that "we suggest", just start the sentence by stating that "Bacterioplankton dynamics also must be better represented in biogeochemical models."**
The sentence was modified as follows: "Bacterioplankton dynamics also must be better represented in biogeochemical models."

**Line 356. Add "such" prior to as dissolved organic carbon**
Done

**Line 359-360. change "would help" to "would be helpful"**
Done

**Figure 1. It is hard to the Mackenzie Bay label on the figure with the color scheme being used. The sub-figures could also be larger and a larger scale map of the Mackenze delta and locating it within a regional map of the Arctic would be helpful.**
The three labels were colored in bold black to improve the readability. A new figure (Fig. 1) was also added to locate the study area within the whole Arctic Ocean. As such, we modified the text (line 114) as follows: "Scene images of tDOC concentrations were used to make monthly composite images at 1 km horizontal resolution of the Mackenzie shelf in the Canadian Beaufort Sea (Fig. 1)."

We gratefully thank **referee #2** for her/his new constructive comments with respect to our manuscript. In order to improve the manuscript with respect to these comments, we amended the manuscript as suggested by the referee wherever it was possible. Note that, when needed, comments were merged together to bring more clarity in the answer:

1. **"The authors use a constant value of 15% bioavailable tDOC for DOC delivered to the Canadian Arctic Ocean by rivers based on Wickland et al 2012 Table 5. (Wickland et al 2012 is missing from Reference list). That value is based on an extrapolation from Yukon river experiments applied to the six largest Arctic rivers. The 15% value is at the low end of what other studies have found, e.g. Holmes et al. 2008 estimated 15-33%, Alling et al 2010 found 30-50% tDOC removal over inner ESAS, Letscher et al 2011; 2013 found 40-60% removal of tDOC and tDON over the ESAS. This underestimate of the bioavailable fraction of tDOC upon delivery to the Arctic Ocean could be the major reason why their simulated values of tDOC are consistently overestimated when compared to satellite estimated tDOC for the outer shelf and offshore locations (Fig 1, Table 1). An underestimation of the tDOC remineralization rate could also drive a similar pattern."**

We modified the text (line 249-263) to account for the reviewers' comment: "In the model, the removal of tDOC through photo-oxidation (Bélanger et al., 2006) was not taken into account. Assuming an annual mean mineralization rate of tDOC of ~0.02 TgC (Bélanger et al., 2006), this process would explain <2 % of the reported tDOC difference in August. In addition, the 15% value used to set the bioavailable tDOC fraction in the model was at the low end of values reported in other studies (up to 50%; Mann et al., 2012; Wickland et al., 2012, Letscher et al., 2011; Alling et al., 2010; Holmes et al., 2008). This underestimation of the bioavailable fraction of tDOC upon delivery to the AO could be a major reason why the simulated values of tDOC were consistently overestimated when compared to satellite estimates for the outer shelf and offshore locations (Fig. 1, Table 1). In the model, bacterioplankton consumed tDOC to produce ammonium usable in turn by phytoplankton. In the Beaufort Sea, this pathway contributed to primary production by 35 % on average over 2003-2011. However, the simulated rates of bacterioplankton production (< 30 mgC $m^{-2}$ $d^{-1}$) still remained in the lower range of those measured in the Beaufort Sea (25-68 mgC $m^{-2}$ $d^{-1}$; Ortega-Retertua et al., 2012; Vallières et al., 2008). The likely underestimation of the tDOC removal by bacterioplankton in the model during summer months might largely contribute to the reported bias between the model and the satellite data."

2. **"The authors should state what tDOC remineralization rate they are using in their model since a few observational based estimates exist in the literature for their study region."**

We cannot provide such simulated data as all parts (i.e. biogeochemical rates) of the partial differential equations were not saved in the model outputs. However, we compared the simulated bacterioplankton production rates with the measured rates available in our study area and in other AO shelf seas as well (see Le Fouest et al., 2015). This is an indirect (but we agree, also incomplete) way to assess the performance of the model in simulating the DOC remineralization rate.

3. **"However, for the computed lateral transport fluxes of tDOC (Fig 3) it is shown that the model overestimates of tDOC concentration don't have a large effect on the computed lateral tDOC transport fluxes (<20%). Thus I don't see a major detriment to the utility of**

**this study and its conclusions related to this consistent model tDOC bias. The authors do point out the likely reasons for this model discrepancy and how targeted field and laboratory studies could help inform model parameterization of tDOC removal in the Arctic Ocean."**

**Major comment:**

**4. Line 251-254. How are the rates of primary production and bacterioplankton production due to tDOC decomposition to ammonium computed?**

The primary production rate based on ammonium is computed as the product of the phytoplankton biomass with the growth rate (computed as the minimum of the light-based and nutrient-based growth rate) and the nutrient limitation term (dimensionless) computed according to the substitutable model of O'Neill et al. (1989).

The way that the bacterioplankton production rate based on ammonium is computed is more complex. It is fully detailed in section A3 of the appendix in Le Fouest et al. (2015). It is the product of the bacterioplankton biomass with the maximum growth rate, the ammonium limitation term (dimensionless), and the temperature limitation term (Q10 fomulation).

**5. How is the source of ammonium (tDOC breakdown vs internal marine recycling) differentiated in the model?**

The biological sources of ammonium in the model are microzooplankton through recycling (egestion process based on bacterioplankton and small phytoplankton grazing), and mesozooplankton through excretion. These processes are identified in tables 1 and 2, and the mechanistic equations detailed in section A2 of the appendix in Le Fouest et al. (2015).

**6. Is there a model sensitivity test performed with a simulation with tDOC cycling turned on and one simulation with no tDOC cycling and the primary/bacterial production rates compared?**

Yes, such a test was performed in the study of Le Fouest et al. (2015).

**7. The authors need to provide more detail on how these estimates are computed from their model outputs.**

We hope we provided the details required by the referee in our answers to points 4 and 5 above.

**Lastly, the Perspectives section provides an accurate telling of the state of the field with regards to being able to model the fate of tDOC and its current limitations. Numerous suggestions are made for where focused observational and laboratory studies can help inform improved tDOC parameterizations in Arctic coupled physical-biogeochemical models.**

Cited references
O'Neill, R. V., DeAngelis, D. L., Pastor, J. J., Jackson, B. J., and Post, W. M.: Multiple nutrient limitations in ecological models, Ecol. Model., 46, 147–163, 1989.

[revised manuscript text omitted]